# Minimax Regret of Switching-Constrained Online Convex Optimization: No Phase Transition

**Lin Chen**[1,2*]   **Qian Yu**[3*]   **Hannah Lawrence**[4]   **Amin Karbasi**[1]
[1] Yale University    [2] Simons Institute for the Theory of Computing
[3] University of Southern California    [4] Massachusetts Institute of Technology

## Abstract

We study the problem of switching-constrained online convex optimization (OCO), where the player has a limited number of opportunities to change her action. While the discrete analog of this online learning task has been studied extensively, previous work in the continuous setting has neither established the minimax rate nor algorithmically achieved it. In this paper, we show that $T$-round switching-constrained OCO with fewer than $K$ switches has a minimax regret of $\Theta(\frac{T}{\sqrt{K}})$. In particular, it is at least $\frac{T}{\sqrt{2K}}$ for one dimension and at least $\frac{T}{\sqrt{K}}$ for higher dimensions. The lower bound in higher dimensions is attained by an orthogonal subspace argument. In one dimension, a novel adversarial strategy yields the lower bound of $O(\frac{T}{\sqrt{K}})$, but a precise minimax analysis including constants is more involved. To establish the tighter one-dimensional result, we introduce the *fugal game* relaxation, whose minimax regret lower bounds that of switching-constrained OCO. We show that the minimax regret of the fugal game is at least $\frac{T}{\sqrt{2K}}$ and thereby establish the optimal minimax lower bound in one dimension. To establish the dimension-independent upper bound, we next show that a mini-batching algorithm provides an $O(\frac{T}{\sqrt{K}})$ upper bound, and therefore conclude that the minimax regret of switching-constrained OCO is $\Theta(\frac{T}{\sqrt{K}})$ for any $K$. This is in sharp contrast to its discrete counterpart, the switching-constrained prediction-from-experts problem, which exhibits a phase transition in minimax regret between the low-switching and high-switching regimes.

## 1   Introduction

Online learning provides a versatile framework for studying a wide range of dynamic optimization problems, with manifold applications in portfolio selection [18], packet routing [7], hyperparameter optimization [19], and spam filtering [23]. The fundamental problem is typically formulated as a repeated game between a player and an adversary. In the $t^{th}$ round, the player first chooses an action $x_t$ from the set of all possible actions $\mathcal{D}$; the adversary then responds by revealing the penalty for that action, a function $f_t : \mathcal{D} \to \mathbb{R}$. The player's goal is to minimize the total penalties she receives, while the adversary's goal is to maximize the penalties she assigns to the player's action. Explicitly, the standard benchmark for success is regret, the difference between the player's accumulated penalty and that of the best fixed action in hindsight: $\mathcal{R} = \sum_{i=1}^{T} f_i(x_i) - \inf_{x \in \mathcal{D}} \sum_{i=1}^{T} f_i(x)$ .

Several variants of this general learning setting have been studied. When $\mathcal{D}$ is a discrete set of actions, the game is called either "prediction from experts" (PFE), if the player is allowed knowledge of the complete function $f_t(\cdot)$ on each round, or "multi-armed bandit" (MAB), if only $f_t(x_t)$ is revealed on each round. Crucially, the adversary is not strongly adaptive and picks $f_t$ based solely on prior

knowledge of the player's randomized strategy and $x_1, \ldots, x_{t-1}$, and not $x_t$; otherwise, she could always force linear regret in $T$ [12, 24]. In this paper, we instead consider the continuous analog of prediction from experts, termed *online convex optimization*. Here, $\mathcal{D}$ is a continuum of possible actions, but the entirety of $f_t(\cdot)$ is revealed after it has been played. Surprisingly, the continuity of $\mathcal{D}$ means that a player can guarantee sublinear regret against a strongly adaptive adversary, i.e., one who may choose $f_t$ even after observing $x_t$.

In many real-world applications, however, we desire an online algorithm to have greater continuity in its actions over the course of many rounds. In caching, for example, erratic online decisions may induce cache misses, and thus costly memory access procedures [10]. More explicitly, the number of times that the player can switch her action between rounds may be strictly constrained. For example, suppose that the player makes predictions based on expert advice. If she would like to hire a new expert, she has to terminate the current contract, pay an early termination fee, and hire and pay a new expert. If hiring a new expert costs $1000 in total and her budget is $10000 dollars, the number of her switches must be less than 10. This setting is called *switching-constrained* or *switching-budgeted* online learning [3]. In these settings, it is necessary to assume an oblivious adversary: an adaptive adversary can force an algorithm with fewer than $K$ switches to incur linear regret by assigning 0 to a switched action between rounds, and 1 to a repeated action [3].

Previous work has established the minimax regret of the switching-constrained multi-armed bandit and prediction from experts problems, but the minimax rate of switching-constrained online convex optimization was neither known nor algorithmically achieved. In this paper, we establish the minimax regret of *switching-constrained* online convex optimization (OCO) against the strongest adaptive adversary, and in doing so, present a simple mini-batching algorithm that achieves this optimal rate.

We assume that $\mathcal{D}$, the action set of the player, is a compact, convex subset of $\mathbb{R}^n$. Let $\mathcal{F}$ be a family of differentiable convex functions from $\mathcal{D}$ to $\mathbb{R}$ from which the adversary selects each round's loss function, $f_t$. In the full-information setting (OCO), we assume that the player observes the loss function $f_t$ after the adversary decides on it. The key ingredient, differentiating our setting from typical OCO, is a limit on the player's number of switches. Formally, given a sequence of points $x_1, \ldots, x_T$, let $c(x_1, \ldots, x_T) = \sum_{i=1}^{T-1} \mathbb{1}[x_{i+1} \neq x_i]$ denote the number of switches. The player's action sequence must satisfy $c(x_1, \ldots, x_T) < K$ for some natural number $K$.[2]

Given the player's action sequence $x_1, \ldots, x_T$ and the adversary's loss sequence $f_1, \ldots, f_T$, the usual regret is defined by the total accumulated loss incurred by the player, minus the total loss of the best possible single action in hindsight. We add an additional term and an outermost supremum that drives the regret of any player's sequence that violates the switch limit to infinity:

$$\mathcal{R}(\{x_i\}, \{f_i\}) = \sup_{\lambda > 0} \left( \sum_{i=1}^{T} f_i(x_i) - \inf_{x \in \mathcal{D}} \sum_{i=1}^{T} f_i(x) + \lambda \mathbb{1}[c(x_1, \ldots, x_T) \geq K] \right),$$

where $\mathbb{1}[\cdot]$ is the statement function whose value is 1 if the proposition inside the brackets holds and is 0 otherwise. In the following sections, we denote the switching-constrained minimax regret by

$$\mathcal{R}(T, K) = \inf_{x_1} \sup_{f_1} \ldots \inf_{x_T} \sup_{f_T} \mathcal{R}(\{x_i\}, \{f_i\}),$$

where it will be clear from context from which sets $x_i$ and $f_i$ may be drawn.

## 2 Related Work

The framework of online convex optimization (OCO) and online gradient descent were introduced by Zinkevich [25]. Abernethy et al. [1] showed that the minimax regret of OCO against a strong adversary is $\Theta(\sqrt{T})$. Abernethy et al. [2] provided a geometric interpretation, demonstrating that the optimal regret can be viewed as the Jensen gap of a concave functional, and McMahan and Abernethy [21] studied the minimax behavior of unconstrained online linear optimization (OLO).

There is a substantial body of literature for switching-constrained and switching-cost online learning in the discrete settings, MAB and PFE. Switching-cost learning forces the player to pay for each

switch rather than enforcing a strict upper bound. Assuming potentially unbounded loss per round, Cesa-Bianchi et al. [11] first showed that the minimax optimal rates of PFE and MAB with switching costs are $\Theta(\sqrt{T})$ and $\tilde{\Theta}(T^{2/3})$, respectively. Dekel et al. [14] proved that the minimax regret of MAB with a unit switch cost in the standard setup (losses are bounded in $[0, 1]$) is $\tilde{\Theta}(T^{2/3})$. Devroye et al. [15] proposed a PFE algorithm whose expected regret and expected number of switches are both $O(\sqrt{T \log n})$, where $n$ is the size of the action set. Finally, Altschuler and Talwar [3] showed that there is a phase transition, with respect to $K$, in switching-constrained PFE. If the maximum number of switches $K$ is $O(\sqrt{T \log n})$ (low-switching regime), the optimal rate is $\min\{\tilde{\Theta}(\frac{T \log n}{K}), T\}$. If $K$ is $\Omega(\sqrt{T \log n})$ (high-switching regime), the optimal rate is $\tilde{\Theta}(\sqrt{T \log n})$. Once at least $\sqrt{T \log n}$ switches are permitted, the minimax regret surprisingly is not improved at all by allowing even more switches. In contrast to PFE, switching-constrained MAB exhibits no phase transition and the minimax rate is $\min\{\tilde{\Theta}(\frac{T \sqrt{n}}{\sqrt{K}}), T\}$.

Within the switching-constrained literature for continuous OCO, most directly comparable to our setting is Jaghargh et al. [17], which proposed a Poisson process-based OCO algorithm. The algorithm's expected regret is $O(\frac{T^{3/2}}{\mathbb{E}[K]})$, where $\mathbb{E}[K]$ may be set to any value provided that $\mathbb{E}[K] = \Omega(\sqrt{T})$. The expected regret, as a function of the expected number of switches, is suboptimal relative to the switching-constrained minimax rate we prove and achieve in this work.

In the related learning with memory paradigm, the loss function for each round depends on the $M$ most recent actions. Switching-cost OCO can be viewed as a special case of learning with memory, in which $M = 1$ and the loss functions are $g(x_t, x_{t-1}) = f(x_t) + c\mathbb{1}[x_t \neq x_{t-1}]$. Merhav et al. [22] introduced the concept of learning with memory, and used a blocking technique to achieve $O(T^{2/3})$ policy regret (a modification of standard regret for adaptive adversaries) and $O(T^{1/3})$ switches against an adaptive adversary. Arora et al. [6] formally clarified and expanded the notion of policy regret for learning with memory, and presented a generalized mini-batching technique (applied here to achieve the matching upper bound) for online bandit learning against an adaptive adversary, converting arbitrary low regret algorithms to low policy regret algorithms. In Appendix A, we also briefly discuss metrical task systems and online optimization with normed switching costs, but the main focus of this paper is the switching-constrained setting.

## 3 Contributions

In this paper, we show that if both the player and the adversary select from the $L_2$ ball (i.e., $||x_t||_2 \leq 1$ and $f_t(x_t) = w_t \cdot x_t$ with $||w_t||_2 \leq 1$), then the minimax regret $\mathcal{R}(K, T)$ of switching-constrained online linear optimization is $\Theta(\frac{T}{\sqrt{K}})$. The precise bounds are contained below.

**Theorem 1** (Minimax regret of OLO). *The minimax regret of switching-constrained online linear optimization satisfies the following bounds:*

*(a)* $\frac{T}{\sqrt{2K}} \leq \mathcal{R}(K, T) \leq \lceil \frac{T}{K} \rceil \sqrt{\frac{2(K+1)}{\pi}} \leq 2\sqrt{\frac{2}{\pi}} \frac{T}{\sqrt{K}}$ *for $n = 1$;*
*(b)* $\frac{T}{\sqrt{K}} \leq \mathcal{R}(K, T) \leq \lceil \frac{T}{K} \rceil \sqrt{K} \leq \frac{2T}{\sqrt{K}}$ *for all $n > 1$.*

In Section 4.1, we prove the lower bound for dimension $n$ greater than 1, and in Section 4.2, we prove a one-dimensional lower bound that is slightly weaker than that of part (a), with prefactor $\frac{1}{2}$. To obtain the one-dimensional lower bound with tight constant $\frac{1}{\sqrt{2}}$, in Section 6 we analyze a carefully chosen OCO relaxation termed the "fugal game". For ease of presentation, we provide an overview of the key intuitions of the fugal game and defer the complete analysis to the Supplementary Materials.

Whereas it was sufficient to assume the adversary chose linear functions in all of the lower bounds, the upper bounds above are in fact derived from a far more general class of convex functions, as shown in the following proposition.

**Proposition 2.** *If $\mathcal{D}$ is a convex and compact set from which the player draws $x_i$, and $\mathcal{F}$ is the family of differentiable convex functions on $\mathcal{D}$, with uniformly bounded gradient, from which the adversary chooses $f_i$, a mini-batching algorithm yields the upper bound $\mathcal{R}(T, K) \leq \lceil \frac{T}{K} \rceil O(\sqrt{K}) = O(\frac{T}{\sqrt{K}})$.*

Building on the previous proposition, the precise constants of the upper bounds in parts (a) and (b) for the case of linear functions are proven in the Supplementary Material (Proposition 32 and

Proposition 34). Combining the results of Theorem 1 and Proposition 2 immediately yields the following key minimax rate as a corollary.

**Corollary 3.** *The minimax regret of $T$-round OCO with fewer than $K$ switches is $\Theta(\frac{T}{\sqrt{K}})$.*

Before proceeding with the proofs of the preceding lower and upper bounds, we first consider a few implications and subtleties of the $O(\frac{T}{\sqrt{K}})$ minimax rate. First, note that if the player is not allowed to make any switch once she decides on her first action ($K = 1$), the minimax regret $\Theta(\frac{T}{\sqrt{K}}) = \Theta(T)$ is, naturally, linear. If the player is allowed to make more than $T - 1$ switches, $\Theta(\frac{T}{\sqrt{K}}) = \Theta(\frac{T}{\sqrt{T}}) = \Theta(\sqrt{T})$ recovers the classical $\Theta(\sqrt{T})$ regret of OCO [1].

Furthermore, this minimax rate is in sharp contrast to switching-constrained PFE, the discrete counterpart of switching-constrained OCO. As noted in Section 2, Altschuler and Talwar [3] proved a phase transition in switching-constrained PFE between the high-switching and low-switching regimes. However, in the continuous full-information (OCO) setting, the minimax regret is the same regardless of the number of switches. [3]

We also evaluate more closely the constants in the lower and upper bounds of Theorem 1 in the Supplementary Material, and mention the key results here.

**Proposition 4** (The constant in $\frac{T}{\sqrt{2K}}$ is unimprovable). *The constant $\frac{1}{\sqrt{2}}$ in the lower bound $\mathcal{R}(T, K) \geq \frac{T}{\sqrt{2K}}$ cannot be increased.*

We prove this by considering the special case $K = 2$ (Appendix C.8). [4]

In addition, the result of Theorem 1 exhibits a similar phenomenon to that observed by [21] in the dimension-dependent minimax behavior of ordinary OCO. McMahan and Abernethy [21] noted that the one-dimensional minimax value is approximately $0.8\sqrt{T}$, while in higher dimensions (where both the player and the adversary select from the $n$-dimensional Euclidean ball) it is exactly $\sqrt{T}$. In a switching-constrained OLO game, if the dimension is greater than 1, Theorem 1 shows that the minimax regret is asymptotically $\frac{T}{\sqrt{K}}$ for all sufficiently large $T$. The following general proposition (proven in Appendix E) provides a link between the regret of the higher-dimensional and one-dimensional games:

**Proposition 5.** *The minimax regret $\mathcal{R}(T, K)$ is non-decreasing in the dimension $n$.*

As a consequence, by part (a) of Theorem 1 and Proposition 5, the one-dimensional minimax regret is asymptotically between $\frac{T}{\sqrt{2K}} \approx 0.7\frac{T}{\sqrt{K}}$ and $\frac{T}{\sqrt{K}}$ for all sufficiently large $T$. In particular, if $K > 1$, we establish further in Appendix E (Proposition 34) that it is at most $0.87\frac{T}{\sqrt{K}}$ for all sufficiently large $T$. This $0.87\frac{T}{\sqrt{K}}$ upper bound is strictly less than the higher-dimensional rate $\frac{T}{\sqrt{K}}$. Thus the constant in the one-dimensional case is distinct from that of all higher dimensions.

## 4 Lower Bound

We separately consider the high-dimensional (Section 4.1) and one-dimensional (Section 4.2) lower bounds, presenting distinct adversarial strategies for each.

### 4.1 Dimension $n > 1$

In this section, we present the primary lower bound result for higher dimensions; a dimension-dependent result, in which player and adversary select from the $L_\infty$ rather than $L_2$ ball, can be found

in Appendix D. We call the first round and every round in which the player chooses a new point a *moving* round, and all other rounds *stationary* rounds.

The adversary's strategy attaining this lower bound is to follow a switching pattern identical to the player's, and to select a point via the orthogonal trick originally introduced in [1]. It was non-trivial to adapt the orthogonal trick to the switching-constrained setting. First, to prove our result, we had to impose a certain switching pattern upon the adversary which was not obvious *a priori*, since the adversary is free to play any functions they wish from round to round. Without constraining the adversary to follow the player's switching pattern, the orthogonal trick cannot be applied. In addition, we found that this adversary's strategy can be adapted for $n = 2$ — thereby avoiding the need for special treatment as in $n = 1$ — via a geometric fact about the intersection of two closed half-spaces. [1] only covered $n > 2$.

By the straightforward calculation $-\inf_{||x||_2 \leq 1} \sum_{i=1}^{T} w_i \cdot x = \sup_{||x||_2 \leq 1} \sum_{i=1}^{T} w_i \cdot x = \left\| \sum_{i=1}^{T} w_i \right\|$, the regret $\mathcal{R}(T, K)$ has two terms, $\sum_{i=1}^{T} w_i \cdot x_i$ and $\left\| \sum_{i=1}^{T} w_i \right\|$. The adversary wishes to make both terms non-decreasing over time. At the $i$-th round, if it is a moving round of the player, the adversary can choose a point $x_i$ whose inner products with $w_i$ and $\sum_{j<i} w_j$ are both non-negative. If it is a stationary round, the adversary selects her previous point.

**Proposition 6** (Lower bound for higher dimensions). *The minimax regret $\mathcal{R}(T, K)$ is at least $\frac{T}{\sqrt{K}}$ for all $n > 1$.*

*Proof.* Let $1 = m_1 < m_2 < \cdots < m_K$ denote all moving rounds, with $m_{K+1} = T + 1$ and $M_i = m_{i+1} - m_i$ denoting the length between two consecutive moving rounds. Also, for any integer $1 \leq t \leq T$, let $\pi(t)$ be the unique integer such that $m_{\pi(t)} \leq t < m_{\pi(t)+1}$, and define $W_t = \mathbb{1}[t > 1] \sum_{j=1}^{t} w_j$. Let us consider this adversary's strategy. At the $t$-th round, if $t$ is a moving round, the adversary chooses $w_t$ such that $\|w_t\| = 1$, $w_t \cdot x_t \geq 0$, and $w_t \cdot W_{t-1} \geq 0$.

Such a vector $w_t$ exists provided that the dimension $n \geq 2$. For $n > 2$, the subspace of $\mathbb{R}^n$ such that the latter two conditions are tight is of dimension $n - 2 \geq 1$ and we may choose $w_t$ from this subspace. For $n = 2$, the latter two conditions each define a closed halfspace of $\mathbb{R}^2$ and thus must have a non-empty intersection. If $t$ is a stationary round, the adversary chooses $w_{m_{\pi(t)}}$, *i.e.*, the same vector that she plays at the moving around. Then the regret becomes

$$\sum_{t=1}^{T} w_t \cdot x_t - \inf_{x \in B_2^n} \sum_{t=1}^{T} w_t \cdot x \geq - \inf_{x \in B_2^n} W_T \cdot x = \|W_T\|.$$

Now let us lower bound $\|W_T\|$. By the choice of $w_{m_i}$, $w_{m_i}$ is perpendicular to $\sum_{j=1}^{i-1} M_j w_{m_j}$. By iterating this relation, we obtain

$$\left\| \sum_{i=1}^{K} M_i w_{m_i} \right\|^2 = \sum_{i=1}^{K} ||M_i w_{m_i}||^2 = \sum_{i=1}^{K} M_i^2 ||w_{m_i}||^2.$$

It is thus the case that

$$\|W_T\| = \left\| \sum_{i=1}^{K} M_i w_{m_i} \right\| \geq \sqrt{\sum_{i=1}^{K} M_i^2 ||w_{m_i}||^2} = \sqrt{\sum_{i=1}^{K} M_i^2} \geq \frac{T}{\sqrt{K}},$$

where the last inequality follows from the Cauchy-Schwarz inequality. $\square$

## 4.2 Dimension $n = 1$

For the one-dimensional case, we present a strategy for the adversary that guarantees regret at least $\frac{T}{2\sqrt{K}}$, the correct order of minimax regret. However, note that the prefactor $\frac{1}{2}$ is suboptimal, and in Section 6 we elaborate on a more involved analysis that will yield a tighter constant.

**Proposition 7** (Lower bound for one dimension). *The minimax regret $\mathcal{R}(T, K)$ is at least $\frac{T}{2\sqrt{K}}$ for $n = 1$.*

*Proof.* Let $W_t = \sum_{i=0}^{t-1} w_i$. Recall from Section 4.1 that the regret consists of two terms, $\sum_i x_i w_i$ and $|\sum_i w_i|$. At a high level, the adversary either chooses to maximize the first term, or determines that the contribution of the second term is high enough and prevents the regret from changing further by selecting the 0 loss function for the remainder of the rounds. Concretely, let $w_t = 0$ if $|W_t| \geq T/\sqrt{K}$. We refer to this as the "stopping condition". When $|W_t| < T/\sqrt{K}$, let $w_t = 1$ if $x_t \geq -W_t \sqrt{K}/T$, and $w_t = -1$ otherwise.

Let $t_i$ denote the round of the $i$th switch in $x$, with $t_0 = 0$, and let $t_K$ denote the round where $|W_t|$ reaches $\frac{T}{\sqrt{K}}$, or $T$ if it does not exist.[5] Let $T_i = t_{i+1} - t_i$ then denote the length of the $(i+1)$st block, and note that $w_t$ remains fixed within each interval, i.e. the adversary copies the player's switching pattern. We then have

$$\mathcal{R}(T, K) = \sum_{t<T} x_t w_t + |W_{t_K}| = \sum_{i=0}^{K-1} (x_{t_i} + W_{t_i} \sqrt{K}/T) w_{t_i} T_i - \sum_{i=0}^{K-1} (W_{t_i} \sqrt{K}/T) w_{t_i} T_i + |W_{t_K}|.$$

By the adversary's strategy, $(x_{t_i} + W_{t_i} \sqrt{K}/T) w_{t_i}$ is always non-negative, and so this expression is at most $-\sum_{i=0}^{K-1} (W_{t_i} \sqrt{K}/T) w_{t_i} T_i + |W_{t_K}|$. Since $w_t$ within each interval is fixed, we have $w_{t_i} T_i = W_{t_{i+1}} - W_{t_i}$. Moreover, it suffices to consider only $i$ in the sum such that $w_i$ has not yet been set to 0. Applying these facts, the expression becomes:

$$-\sum_{i=0}^{K-1} \left(\frac{\sqrt{K}}{2T}\right) (W_{t_{i+1}}^2 - W_{t_i}^2 - (w_{t_i} T_i)^2) + |W_{t_K}|$$

$$= -\left(\frac{\sqrt{K}}{2T}\right) W_{t_K}^2 + \sum_{i=0}^{K-1} \left(\frac{\sqrt{K}}{2T}\right) T_i^2 + |W_{t_K}|.$$

Note that $\sum_{i=0}^{K-1} T_i = t_K$, so by Cauchy-Schwarz we can lower bound this expression by $-\left(\frac{\sqrt{K}}{2T}\right) W_{t_K}^2 + \left(\frac{1}{2T\sqrt{K}}\right) t_K^2 + |W_{t_K}|$. If the stopping condition is never met, then $t_K = T$ and the remaining terms are non-negative, since they factorize as $|W_{t_K}|(1 - \frac{\sqrt{K}}{2T} W_{t_K})$ and $|W_{t_K}| < \frac{T}{\sqrt{K}}$ by assumption. Otherwise, we have $|W_{t_K}| \in [\frac{T}{\sqrt{K}}, \frac{T}{\sqrt{K}}+1]$, and thus $(|W_{t_K}| - \frac{T}{\sqrt{K}})(|W_{t_K}| - \frac{T}{\sqrt{K}} - 1) \leq 0$. Consequently, the chain of lower bounds continues with:

$$\mathcal{R}(T, K) \geq \min\left\{ \frac{T^2}{2T\sqrt{K}}, -\left(\frac{\sqrt{K}}{2T}\right) W_{t_K}^2 + \frac{t_K^2}{2T\sqrt{K}} + |W_{t_K}| \right.$$

$$\left. + \left(\frac{\sqrt{K}}{2T}\right)\left(|W_{t_K}| - \frac{T}{\sqrt{K}}\right)\left(|W_{t_K}| - \frac{T}{\sqrt{K}} - 1\right) \right\}.$$

Finally, this entire expression is lower bounded by $\frac{T}{2\sqrt{K}}$. This follows by noting that the second term equals $(\frac{1}{2T\sqrt{K}}) t_K^2 + |W_{t_K}|(-\frac{\sqrt{K}}{2T}) + (\frac{T}{\sqrt{K}} + 1)/2$, which is greater than or equal to

$$\left(\frac{1}{2T\sqrt{K}}\right) t_K^2 \left(\frac{T}{\sqrt{K}} + 1\right)\left(-\frac{\sqrt{K}}{2T}\right) + \left(\frac{T}{\sqrt{K}} + 1\right)/2 = \left(\frac{1}{2T\sqrt{K}}\right) t_K^2 - \frac{\sqrt{K}}{2T} + \frac{T}{2\sqrt{K}}.$$

Noting that the second term assumed the stopping condition was met and therefore $t_K \geq |W_{t_K}| \geq \frac{T}{\sqrt{K}}$, the resulting quantity is then lower bounded by $(\frac{1}{2T\sqrt{K}})(\frac{T}{\sqrt{K}})^2 - \frac{\sqrt{K}}{2T} + (\frac{T}{\sqrt{K}})/2 \geq (\frac{T}{\sqrt{K}})/2$. $\square$

## 5  Upper Bound

In this section, we prove Proposition 2 and thereby derive an upper bound for switching-constrained OCO to match the lower bounds of Section 4.2 and Section 4.1. We begin with a simple algorithm

achieving the correct minimax regret, $O(\frac{T}{\sqrt{K}})$. In Appendix E, we expand on this result with a closer evaluation of the constant.

*Proof.* First, we claim that the minimax regret $\mathcal{R}(T, K)$ is a non-decreasing function in $T$. To see this, consider the situation where we have more rounds. The adversary can play $0$ in all additional rounds and this does not decrease the regret. Therefore, we obtain that $\mathcal{R}(T, K) \leq \mathcal{R}(T_1, K)$, where $T_1 = \lceil \frac{T}{K} \rceil K \geq T$.

To derive an upper bound for $\mathcal{R}(T_1, K)$, we mini-batch the $T_1$ rounds into $K$ equisized epochs, each having size $\frac{T_1}{K} = \lceil \frac{T}{K} \rceil$. Let $E_i$ denote the set of all rounds that belong to the $i$-th epoch. We have $E_i = \{\frac{T_1}{K}(i-1)+1, \frac{T_1}{K}(i-1)+2, \ldots, \frac{T_1}{K}i\}$. The epoch loss of the $i$-th epoch is the average of loss functions in this epoch, *i.e.*, $\bar{f}_i \triangleq \frac{1}{|E_i|} \sum_{j \in E_i} f_j$. If we run a minimax optimal algorithm for unconstrained OCO (for example, online gradient descent [25]) on the epoch losses $\bar{f}_1, \ldots, \bar{f}_K$ and obtain the player's action sequence $\bar{x}_1, \ldots, \bar{x}_K$, our strategy is to play $\bar{x}_i$ at all rounds in the $i$-th epoch. This method was originally discussed in [6, 13]. Using this mini-batching method, we deduce that the regret is upper bounded by $\frac{T_1}{K}O(\sqrt{K}) = \lceil \frac{T}{K} \rceil O(\sqrt{K}) = O(\frac{T}{\sqrt{K}})$, where $O(\sqrt{K})$ is the standard upper bound of the regret of a $K$-round OCO game. $\qquad\square$

# 6 Exact Minimax Bound

In Section 4.2, we analyzed an adversarial strategy for OLO that gave the desired order of lower bound, $O(\frac{T}{\sqrt{K}})$. However, a separate (and far more involved) analysis is needed to obtain the tight constant in $\frac{T}{\sqrt{2K}}$, which revealed the surprising parallels (in the differing constants of one and higher dimensions) between switching constrained and unconstrained OCO that were stated in Section 3. In this section, we outline the key ideas of our approach to obtaining a *tight* lower bound. The analysis to complete the proof of this result is in Appendix C.

The reader may wonder why a simpler argument cannot prove the tight one-dimensional lower bound. To shed some light on this difficulty, we note that the truly minimax optimal strategy does not necessarily involve the player choosing to switch at uniform points throughout the $T$ rounds. In Proposition 36 of the Supplementary Material, we consider the case where $K = 3$ and prove that the first switch happens at round approximately $0.29T$, rather than $0.33T$. Although the mini-batching algorithm of our upper bound makes uniformly spaced switches, this simple fact demonstrates that the resultant constant cannot be tight, and that such an assumption — simplifying though it may be — would not be valid for the lower bound.

## 6.1 Fugal Game

Our minimax analysis for the one-dimensional game (OCO) is through what we call *fugal games*. In a fugal game, the adversary is weakened by being constrained to adhere to the player's switching pattern, and to only select from $-1$ and $1$. Furthermore, the horizon $T$ and the interval between consecutive switches are allowed to take non-negative real values; one way of interpreting this is that we allow the player to switch in the "middle" of a round, provided they still respect the switching budget overall and play for a total of $T$ complete rounds. Note that, because the adversary is forced to maintain the same action until the player switches, allowing non-negative real-valued interval between consecutive switches only strengthens the player. **The minimax value of fugal games thus provides a lower bound for the minimax value of switching-constrained OCO.**

As motivation for our terminology, recall that at the exposition of a fugue, one voice begins by introducing a particular melodic theme. Afterward, a new voice repeats the same melody for the same duration, but transposed to a new key. This may repeat multiple times as subsequent voices alternate between the introduction of a new melody (sometimes termed the "question"), and its transposed repetition (the "answer"). In the original switching-constrained OCO framework, the adversary is under no obligation to repeat her loss function for the same number of rounds as the player sticks to the same action. However, if we restrict the adversary to copy the switching pattern of the player, their interaction becomes reminiscent of a fugal exposition. The player begins by choosing a key ($x_i$) for her melody, and the adversary necessarily responds at a new pitch ($f_i$); optionally based on $f_i$, the player chooses the duration $M_i$ to maintain pitch $x_i$, and the adversary imitates her by playing $f_i$

for length $M_i$ as well. This repeats until all $K$ question-and-answer pairs are done. Thus, we call this relaxation of OCO the *fugal game*.

**Proposition 8** (Asymptotic tightness of fugal game; informal)**.** *The minimax regret of the fugal game is* equal *to the normalized minimax regret of switching-constrained online convex optimization asymptotically, when $K$ is constant and $T$ approaches infinity.*

The proof of this proposition can be found in the Supplementary Material (Proposition 29), and proceeds by devising a strategy for the player in OCO based on the fugal game. Proposition 8 justifies that the lower bound of the fugal game's minimax regret is the "right" quantity to study, as it captures the correct asymptotic behavior.

**Proof Overview.**  We solve the minimax behavior of fugal games by studying a generalization of their minimax regret function, with an initial bias. This generalization is called a *fugal function*. We first derive the recursive relation of the fugal function, and then show that the fugal function is at least the absolute value of the initial bias. To average out the influence of $T$, we define the *normalized minimax regret* and show that it is indeed independent of $T$. The normalized minimax regret inherits a recursive relation from the fugal function. However, it is mathematically challenging to solve the exact values of the normalized minimax regret. In light of this, we consider an alternative quadratic lower bound whose recursive relation can be solved in closed form, although significant technical effort is required. This constitutes the most computationally "hardcore" section of this paper. Our minimax analysis for the one-dimensional game follows immediately from the quadratic lower bound.

To aid in the recursive analysis, we generalize ordinary minimax regret slightly by introducing an initial bias. Let $I = [-1, 1]$. Formally, the minimax regret with $T$ rounds, a maximum number of $k$ switches, and an initial bias $Z$ is defined by

$$R_k(T, Z) = \inf_{x_1 \in I} \sup_{w_1 \in I} \ldots \inf_{x_T \in I} \sup_{w_T \in I} \sup_{\lambda > 0} \left( \sum_{i=1}^{T} w_i x_i + \left| Z + \sum_{i=1}^{T} w_i \right| + \lambda \mathbb{1} \left[ c(x_1, \ldots, x_T) \geq k \right] \right).$$

We motivate the initial bias $Z$ as follows. When the adversary tries to maximize regret in any given round, her choice is determined by the tradeoff between maximizing the first term and maximizing the second term above. To focus wholly on the first term, the adversary could specifically penalize the player's last action by playing $w_t = \text{sign}(x_t)$. To focus wholly on the second term, the adversary could instead amplify the term within the absolute value by playing $w_t = \text{sign}(\sum_{i=1}^{t-1} w_i)$. At each round, the adversary chooses a value to optimize this tradeoff given the results of previous rounds. When setting up recursive relations between $R_k$ and $R_{k+1}$, the first term decouples neatly by round, but the second term does not. Thus, an initial bias term is necessary for deriving a recursive relation, as a sort of state that is passed between $R_k$'s. Extending this definition to the fugal game yields the *fugal function*

$$r_k(T, Z) = \inf_{x_1 \in I} \max_{w_1 = \pm 1} \inf_{M_1 \geq 0} \ldots \inf_{x_k \in I} \max_{w_k = \pm 1} \inf_{M_k \geq 0} \sup_{\lambda > 0}$$
$$\left( \sum_{i=1}^{k} M_i w_i x_i + \left| Z + \sum_{i=1}^{k} M_i w_i \right| + \lambda \mathbb{1} [\sum_{i=1}^{k} M_i \neq T] \right),$$

where $M_i$ is the length between two moving rounds, and we have relaxed $M_i$ by allowing it to take any non-negative real values.

We can normalize out the influence of $T$ by setting $u_k(z) = \frac{r_k(T, zT)}{T}$, thereby reducing our task to the analysis of the single-variate function $u_k(z)$. In this way, the fugal game decouples the minimax regret from the discrete nature of $T$. Central to the recursive analysis of $u_k(z)$ is the "fugal operator":

**Definition 1** (Fugal operator)**.** The *fugal operator* transforms continous functions, and is defined by

$$(\mathcal{T} f)(z) \triangleq \inf_{x \in [-1, 1]} \max_{w = \pm 1} \inf_{\substack{|z'| < 1 \\ w(z' - z) \geq 0}} \frac{(1 + wz) f(z') + x(z' - z)}{1 + z'w}.$$

It turns out that $u_k$ satisfies the concise recursive relation $u_{k+1} = \mathcal{T} u_k$. By closely analyzing this relation (the full details of which are contained in Appendix C of the Supplementary Material), we eventually lower bound $u_k(0)$ to obtain a sharp one-dimensional lower bound, $\frac{T}{\sqrt{2K}}$.

# 7 Conclusion

In this work, we considered switching-constrained online convex optimization, a setting which until now had received comparatively little attention relative to switching-constrained multi-armed bandits and prediction from experts. In the OCO setting, we established the minimax regret against the strongest adaptive adversary as $\Theta(\frac{T}{\sqrt{K}})$. The upper bound on minimax regret was constructive, using the mini-batching paradigm to obtain a meta-algorithm for achieving the correct minimax rate. This effectively solves the question of optimal algorithms for switching-constrained online convex optimization.

## Broader Impacts

In this paper, we fully characterize the minimax regret of switching-constrained online convex optimization. Since it is a theoretical result in nature, the broader impact discussion is not applicable.

## Acknowledgments

LC and QY were supported by Google PhD Fellowship. HL was supported by the Fannie and John Hertz Foundation. AK was partially supported by NSF (IIS-1845032), ONR (N00014-19-1-2406), AFOSR (FA9550-18-1-0160), and TATA Sons Private Limited. We would like to thank Jacob Abernethy, Hossein Esfandiari, Peng Zhang, and Peilin Zhong for helpful conversations during the early stages of this work.

## Footnotes

*First two authors contributed equally. Correspondence to: Lin Chen <lin.chen@berkeley.edu>.

[2]In a $T$-round game, the maximum number of switches is always smaller than $T$. As a result, if $K > T$, the game becomes switching-unconstrained. Therefore, we assume throughout this paper that $K \leq T$.

[3]Note that the differing adversary strength is the direct cause of the disparity in phase transitions, but in an online learning problem, one usually assumes the strongest adversary that yields a sublinear minimax regret. The continuity of the (convex) action space circumvents the $\Omega(T)$ regret lower bound [3], allowing sublinear minimax regret against an adaptive adversary, and as such it is the discrete vs. continuous action space that determines the most appropriate adversary.

[4]To clarify, the proven rate $\mathcal{R}(T, K) = \Theta(\frac{T}{\sqrt{K}})$ holds for any $K = o(T)$. For the special case of constant $K$, the exact value of $c$ in $\mathcal{R}(T, K) = c\frac{T}{\sqrt{K}}$ varies by $K$. We concretely compute several examples in Appendix C.6 and then in Appendix C.8 show that for any universal lower bound $\mathcal{R}(T, K) \geq c\frac{T}{\sqrt{K}}$, $c \geq \frac{1}{\sqrt{2}}$.

[5]If the player chooses to use $S < K - 1$ switches, it suffices to assign $\{t_i\}_{i=0}^K$ by arbitrarily splitting up blocks in which the player plays a fixed action, as if she had switched. For example, if the player never switches but $K > 1$, $\{t_i\}$ can be any strictly increasing sequence of round indices.

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
