[Supplementary Material]

# Supplementary Material

## A  Additional Related Work

Metrical task systems is another broad area that overlaps switching-cost OCO, in which the goal is to minimize both movement and cost per round. However, it fundamentally departs from OCO in that the adversary reveals the loss function *first* in each round, and that success is measured by competitive ratio rather than regret. Andrew et al. [5] considered OCO with a seminorm switching penalty added to the regret, and bridged these two modes by demonstrating that no algorithm can simultaneously achieve both sublinear regret and constant competitive ratio. Along the way, they also showed that gradient descent achieves $O(\sqrt{T})$ regret, even with added seminorm switching costs. (Note, however, that a binary penalization for switching is not a seminorm.)

We here mention assorted results in switching-cost or switching-constrained OCO, although they differ significantly in conventions. Instead of a binary penalization for switching per round, Li et al. [20] added a quadratic switching cost to the regret, $\sum_{i=1}^{T-1} \|x_{i+1} - x_i\|^2$. However, they allowed the player some clairvoyance about future loss functions in the form of a fixed "lookahead" window, and thus consider a modified "dynamic" regret. Badiei et al. [9] similarly considered a non-binary switching penalization and a finite lookahead window, with a hard constraint on the total $L_1$ distance between consecutive actions and performance evaluation in terms of the competitive ratio. Gofer [16] demonstrated that for OCO with linear objectives and any normed switching cost, no algorithm can achieve bounded regret against every loss sequence with a finite quadratic variation. Anava et al. [4] presented algorithms for OCO with memory against an *oblivious* adversary, achieving $\tilde{O}(T^{1/2})$ regret and $\tilde{O}(T^{1/2})$ binary switching cost. This result demonstrated that restricting the adversary can lead to regret-switching dependencies stronger than we prove are optimal against an *adaptive* adversary. Awerbuch et al. [8] analyzed limited switching from a multiplicative ratio perspective.

## B  Preliminaries

We denote the $p$-norm by $\|\cdot\|_p$. If $x$ and $y$ are two vectors living in $\mathbb{R}^n$, we write $x \cdot y$ for their inner product. If $x_i$ is a vector, let $x_{i,j}$ denote its $j$-th coordinate. Let $\mathbb{1}[\cdot]$ denote the indicator function whose value is $1$ if the statement inside the brackets holds and is $0$ otherwise.

In the special case of switching-constrained online convex optimization that we focus on, the regret is given by

$$\sum_{i=1}^{T} w_i x_i - \inf_{\|x\|_p \leq 1} \sum_{i=1}^{T} w_i \cdot x = \sum_{i=1}^{T} w_i x_i + \sup_{\|x\|_p \leq 1} \sum_{i=1}^{T} w_i \cdot (x) \stackrel{(a)}{=} \sum_{i=1}^{T} w_i x_i + \left\| \sum_{i=1}^{T} w_i \right\|_{p/(p-1)}. \quad (1)$$

where $(a)$ is because $\|\cdot\|_{p/(p-1)}$ is the dual norm of $\|\cdot\|_p$. Recall that if $\|\cdot\|$ is a norm, its dual norm $\|\cdot\|_*$ is defined by $\|z\|_* \triangleq \sup_{\|x\| \leq 1} z \cdot x$. Let $B_p^n \triangleq \{x \in \mathbb{R}^n : \|x\|_p \leq 1\}$ be the $n$-dimensional unit ball and let $B_q^{*n} = \{f(x) = w \cdot x : w \in \mathbb{R}^n, \|w\|_q \leq 1\}$ denote the dual unit ball, where $1 \leq p, q \leq \infty$. Since all $p$-norms coincide if $n = 1$, we simply write $B^1$ and $B^{*1}$ and do not specify an explicit $p$ and $q$. We also use the more detailed notation $\mathcal{R}_{\mathrm{OCO}}(\mathcal{D}, \mathcal{F}, K, T)$ to indicate that the player chooses actions from $\mathcal{D}$, the adversary chooses functions from class $\mathcal{F}$, and there are $T$ rounds with fewer than $K$ switches.

We use the abbreviations OLO for online linear optimization, OCO for online convex optimization, BLO for bandit linear optimization, and BCO for bandit convex optimization.

**Moving and Stationary Rounds**  We call the first round and every round in which the player chooses a new point a *moving* round. Formally the $i$-th round is a moving round if $i = 1$ or $x_i \neq x_{i-1}$. We call every round in which the player sticks to her previous point a *stationary* round.

# C  Lower Bound for One-Dimensional Switching-Constrained OCO

In this section, we will show the $\frac{T}{\sqrt{2K}}$ lower bound for the minimax regret of the one-dimensional game.

**Proposition 9** (Lower bound for one-dimensional game)**.** *The minimax regret* $\mathcal{R}_{\mathrm{OCO}}(B^1, B^{*1}, K, T)$ *is at least* $\frac{T}{\sqrt{2K}}$*.*

The proof of Proposition 9 can be found in Appendix C.7. However, the proof relies on results in all preceding subsections.

Recall that we defined the minimax regret with $T$ rounds, a maximum number of $k$ switches, and an initial bias $Z$ is by

$$R_k(T, Z) = \inf_{x_1 \in [-1,1]} \sup_{w_1 \in [-1,1]} \ldots \inf_{x_T \in [-1,1]} \sup_{w_T \in [-1,1]} \sup_{\lambda > 0}$$
$$\left( \sum_{i=1}^{T} w_i x_i + \left| Z + \sum_{i=1}^{T} w_i \right| + \lambda \mathbb{1}[c(x_1, \ldots, x_T) \geq k] \right).$$

## C.1  Lower Bound via Fugal Games

As in the main body of the paper, we consider the modified *fugal game*, with the minimax regret of a fugal game with $T$ rounds ($T \in \mathbb{R}_{\geq 0}$), a maximum number of $k - 1$ switches, and an initial bias $Z$ defined by

$$r_k(T, Z) = \inf_{x_1 \in [-1,1]} \max_{w_1 = \pm 1} \inf_{M_1 \geq 0} \ldots \inf_{x_k \in [-1,1]} \max_{w_k = \pm 1} \inf_{M_k \geq 0} \sup_{\lambda > 0}$$
$$\left( \sum_{i=1}^{k} M_i w_i x_i + \left| Z + \sum_{i=1}^{k} M_i w_i \right| + \lambda \mathbb{1}[\sum_{i=1}^{k} M_i \neq T] \right), \tag{2}$$

where $M_i$ is the length between two moving rounds, and we have relaxed $M_i$ by allowing it to take any non-negative real values. The function $r_k(T, Z)$ is a *fugal function*.

The minimax regret in a fugal game gives a lower bound for the minimax regret of interest.[6] In other words, it holds that $r_k(T, Z) \leq R_k(T, Z)$ if $T$ is a natural number. Furthermore, whenever it is the player's turn to play, she must optimize over not only the action, but also the optimal length of time to *maintain* that action to minimize her ultimate regret. As a result of this basic intuition, the function $r_k(T, Z)$ satisfies the following recurrence relation for all $k \geq 1$

$$r_{k+1}(T, Z) = \inf_{x \in [-1,1]} \max_{w = \pm 1} \inf_{0 \leq t \leq T} (twx + r_k(T - t, Z + tw)). \tag{3}$$

## C.2  Absolute Value Bounds for Fugal Games

In this subsection, we derive basic properties of the fugal functions. Lemma 10 shows that the function $r_k(T, Z)$ is at least $|Z|$ for all $Z \in \mathbb{R}$ and that the inequality is tight if $|Z| \geq T$.

**Lemma 10** (Absolute value bounds for fugal games)**.** *The minimax regret of a fugal game with an initial bias* $r_k(T, Z)$ *satisfies the following two properties*

*(a)* $r_k(T, Z) \geq |Z|$ *for all* $Z \in \mathbb{R}$*; and*
*(b)* $r_k(T, Z) = |Z|$ *if* $|Z| \geq T$*.*

*Proof.* To prove part (a), it suffices to design an adversary's strategy that satisfies this lower bound. Suppose that the adversary always plays $\operatorname{sign}(Z)$, or 1 if $Z = 0$. Since $\sum_{i=1}^{k} M_i w_i x_i \geq -\sum_{i=1}^{k} M_i = -T$, if $z \neq 0$ we have

$$\sum_{i=1}^{k} M_i w_i x_i + \left| Z + \sum_{i=1}^{k} M_i w_i \right| \geq -T + |Z + T \operatorname{sign}(Z)| = -T + |(|Z| + T) \operatorname{sign}(z)|$$
$$= -T + (|Z| + T) = |Z| \,.$$

When $Z = 0$, the expression above is clearly at least $-T + T = 0 = |Z|$. Therefore the lower bound $r_k(T, Z) \geq |Z|$ holds.

To prove part (b), we will show that $r_k(T, Z) \leq |Z|$ if $|Z| \geq T$. First, we assume $Z \geq T$. In this case, we have $Z + \sum_{i=1}^{k} M_i w_i \geq T - T = 0$. Since we are certain of the sign of the expression inside the absolute value, we remove the absolute value and obtain

$$\sum_{i=1}^{k} M_i w_i x_i + \left| Z + \sum_{i=1}^{k} M_i w_i \right| = Z + \sum_{i=1}^{k} M_i w_i (1 + x_i) \,.$$

The above expression equals $Z$ if the player always plays $-1$. Therefore, if $Z \geq T$, we obtain $r_k(T, Z) \leq Z = |Z|$.

If $Z \leq -T$, since $Z + \sum_{i=1}^{k} M_i w_i \leq -T + T = 0$, we have

$$\sum_{i=1}^{k} M_i w_i x_i + \left| Z + \sum_{i=1}^{k} M_i w_i \right| = \sum_{i=1}^{k} M_i w_i x_i - (Z + \sum_{i=1}^{k} M_i w_i) = -Z + \sum_{i=1}^{k} M_i w_i (x_i - 1) \,.$$

Again, the above expression equals $-Z$ if the player always plays 1. Therefore, if $Z \leq -T$, we get $r_k(T, Z) \leq -Z = |Z|$. In both cases, we show that $r_k(T, Z) \leq |Z|$, which completes the proof. $\square$

## C.3 Extraspherical Minimax Regret

Recall that in (3), the minimum is taken over all $t$ between 0 and $T$. Let us consider a subset of $t$ such that $|Z + tw| \leq T - t$. Notice that if $t < T$, $|Z + tw|$ belongs to a one-dimensional ball $[-T + t, T - t]$. Lemma 11 gives a minimax lower bound for $t$ such that $|Z + tw|$ is outside the ball. Since it is shown in Lemma 10 that $r_k(T, Z) = |Z|$ if $|Z| \geq T$, we assume in the following lemma that $|Z| < T$.

**Lemma 11** (Extraspherical minimax regret). *If $Z \in (-T, T)$,*

$$\inf_{x \in [-1,1]} \max_{w = \pm 1} \inf_{\substack{0 \leq t \leq T \\ |Z + tw| \geq T - t}} (twx + r_k(T - t, Z + tw)) = \frac{Z^2 + T^2}{2T} \,.$$

*Proof.* Let us first expand the maximum operator

$$A \triangleq \inf_{x \in [-1,1]} \max_{w = \pm 1} \inf_{\substack{0 \leq t \leq T \\ |Z + tw| \geq T - t}} (twx + r_k(T - t, Z + tw))$$

$$= \inf_{x \in [-1,1]} \max \left\{ \inf_{\substack{0 \leq t \leq T \\ |Z + t| \geq T - t}} (tx + r_k(T - t, Z + t)), \inf_{\substack{0 \leq t \leq T \\ |Z - t| \geq T - t}} (-tx + r_k(T - t, Z - t)) \right\} \,.$$

As the first step, we study the situation where $w = 1$. If $|Z + t| \geq T - t$, it implies $Z + t \geq T - t$ or $Z + t \leq -(T - t)$. The second case is impossible since it is equivalent to $Z \leq -T$ (recall our assumption that $|Z| < T$). The first case is equivalent to $t \geq (T - Z)/2$. Therefore the range of $t$ over which the minimum is taken is $\frac{T - Z}{2} \leq t \leq T$. The expression of which we take the infimum becomes

$$tx + r_k(T - t, Z + t) \overset{(a)}{=} tx + |Z + t| = t(x + 1) + Z \,,$$

where $(a)$ is due to Lemma 10 by recalling $|Z + t| \geq T - t$. In this way, since $1 + x \geq 0$, we obtain a cleaner expression for the innermost infimum in the case $w = 1$

$$\inf_{\substack{0 \leq t \leq T \\ |Z+t| \geq T-t}} (tx + r_k(T - t, Z + t)) = \inf_{\frac{T-Z}{2} \leq t \leq T} t(x + 1) + Z = \frac{T - Z}{2}(1 + x) + Z \, .$$

The second step is to study the situation where $w = -1$. If $|Z - t| \geq T - t$, it implies $Z - t \geq T - t$ or $Z - t \leq -(T - t)$. The first case is impossible as it is equivalent to $z \geq 1$. The second case is equivalent to $t \geq (T + Z)/2$. Therefore the range of $t$ over which the minimum is taken becomes $\frac{T+Z}{2} \leq t \leq T$. The expression of which we take the infimum becomes

$$-tx + r_k(T - t, Z - t) = -tx + |Z - t| = t(1 - x) - Z \, .$$

where we use Lemma 10 again in the first equality. Since $1 - x \geq 0$, we obtain a similar clean expression for the innermost infimum in the case $w = -1$

$$\inf_{\substack{0 \leq t \leq T \\ |Z-t| \geq T-t}} (-tx + r_k(T - t, Z - t)) = \inf_{\frac{T+Z}{2} \leq t \leq T} t(1 - x) - Z = \frac{T + Z}{2}(1 - x) - Z \, .$$

Therefore, the extraspherical minimax can be lower bounded as follows

$$A = \inf_{x \in [-1,1]} \max \left\{ \frac{T - Z}{2}(1 + x) + Z, \frac{T + Z}{2}(1 - x) - Z \right\} \, .$$

The first term in the max is greater than the second term if and only if $Tx > -Z$. Therefore, the infimum is attained at $x = -Z/T$ and the extraspherical minimax equals

$$A = \frac{Z^2 + T^2}{2T} \, ,$$

as promised. $\qquad \square$

**Corollary 12.** *If $|Z| < T$, the following recursive relation holds*

$$r_{k+1}(T, Z) = \min \left\{ \inf_{x \in [-1,1]} \max_{w=\pm 1} \inf_{\substack{0 \leq t \leq T \\ |Z+tw| < T-t}} (twx + r_k(T - t, Z + tw)), \frac{Z^2 + T^2}{2T} \right\} \, . \quad (4)$$

Note that if $|Z + tw| < T - t$, it excludes the possibility of $T - t = 0$. Because $T - t > |Z + tw| \geq 0$. We define the normalized regret

$$u'_k(T, z) \triangleq \frac{1}{T} r_k(T, Tz)$$

for $T > 0$ (In the rest of this subsection, we assume $|z| < 1$). Plugging this definition into (4) yields

$$u'_{k+1}(T, z)$$
$$= \min \left\{ \inf_{x \in [-1,1]} \max_{w=\pm 1} \inf_{\substack{0 \leq t < T \\ |Tz+tw| < T-t}} \left( \frac{twx}{T} + (1 - \frac{t}{T})u'_k(T - t, \frac{Tz + tw}{T - t}) \right), \frac{z^2 + 1}{2} \right\} \, .$$

For $0 \leq t < T$ and $|Tz + tw| < T - t$, we define a reparametrization $z'(t) \triangleq \frac{Tz + tw}{T - t}$. The derivative of $z'$ is $\frac{dz'}{dt} = \frac{T(w+z)}{(T-t)^2}$. Since $|z| < 1$, $z'(t)$ is an increasing function if $w = 1$ and is a decreasing function if $w = -1$. If $w = 1$, the system of inequalities $0 \leq t < T$ and $|Tz + t| < T - t$ is equivalent to $0 \leq t < \frac{T(1-z)}{2}$ and thereby $z \leq z' < 1$. If $w = -1$, the system of inequalities $0 \leq t < T$ and $|Tz - t| < T - t$ is equivalent to $0 \leq t < \frac{T(1+z)}{2}$ and thereby $-1 < z' \leq z$. Combining these two cases, we obtain that $|z'| < 1$ and $w(z' - z) \geq 0$.

The definition of $z'$ gives $t = \frac{T(z'-z)}{w+z'}$ and

$$\frac{twx}{T} + (1 - \frac{t}{T})u'_k(T - t, \frac{Tz + tw}{T - t}) = \frac{(w + z)u'_k\left(\frac{T(w+z)}{w+z'}, z'\right) + wx(z' - z)}{w + z'}$$
$$\overset{(a)}{=} \frac{(1 + wz)u'_k\left(\frac{T(w+z)}{w+z'}, z'\right) + x(z' - z)}{1 + z'w} \, ,$$

where $(a)$ is because we multiply the numerator and denominator by $w$ and use the fact that $w = \pm 1$. Therefore, we obtain the following corollary.

**Corollary 13.** *If $|z| < 1$, the following recursive relation holds*

$$u'_{k+1}(T,z) = \min \left\{ \inf_{\substack{x \in [-1,1]}} \max_{w = \pm 1} \inf_{\substack{|z'| < 1 \\ w(z'-z) \geq 0}} \frac{(1+wz)u'_k\left(\frac{T(w+z)}{w+z'}, z'\right) + x\,(z'-z)}{1 + z'w}, \frac{z^2+1}{2} \right\}.$$

$$\tag{5}$$

**Remark 1.** Since both $z$ and $z'$ resides in the open interval $(-1, 1)$ and $w$ is either $-1$ or $1$, the quantity $\frac{w+z}{w+z'}$ is always positive.

### C.4  Normalized Minimax Regret

To derive a closed-form lower bound for $u_k(z)$, we study the boundary condition when $k = 1$.

**Lemma 14** (Boundary condition for $r_1$)**.** *The boundary condition of $r_k(T, Z)$ when $k = 1$ is given by*

$$r_1(T, Z) = \frac{|Z - T| + |Z + T|}{2}.$$

*Proof.* It can be computed directly as below

$$r_1(T, Z) = \inf_{x \in [-1,1]} \max_{w = \pm 1} (Twx + |Z + Tw|).$$

We can expand the innermost maximum, which is minimized at $x = \frac{|z-1| - |z+1|}{2}$, as follows

$$\inf_{x \in [-1,1]} \max_{w = \pm 1} (Twx + |Z + Tw|) = \inf_{x \in [-1,1]} \max(Tx + |Z + T|, -Tx + |Z - T|) = \frac{|z-1| + |z+1|}{2}.$$

Thus, $r_1(T, Z) = \frac{|Z - T| + |Z + T|}{2}$ as claimed. $\qquad\square$

**Corollary 15** (Boundary condition for $u'_1$)**.** *If $|z| < 1$, we have*

$$u'_1(T, z) = \frac{r_1(T, Tz)}{T} = \frac{|z-1| + |z+1|}{2} = 1.$$

**Lemma 16.** *The normalized regret function $u'_k(T, z)$ does not depend on $T$. In other words, there exists a function $u_k(z)$ such that for all $z \in \mathbb{R}$ and $T > 0$, $u'_k(T, z) = u_k(z)$.*

*Proof.* If $|z| \geq 1$, Lemma 10 implies that $u'_k(T, z) = \frac{1}{T} r_k(T, Tz) = \frac{|Tz|}{T} = |z|$, and thereby we define $u_k(z) = |z|$ for $|z| \geq 1$.

If $|z| < 1$, we will prove this lemma by induction on $k$. If we define $u_1(z) = 1$, Corollary 15 shows that $u'_1(T, z) = u_1(z)$. Now we assume that $u'_i(T, z) = u_i(z)$ holds for $i \geq 1$. By Corollary 13, we have

$$u'_{i+1}(T,z) = \min \left\{ \inf_{\substack{x \in [-1,1]}} \max_{w = \pm 1} \inf_{\substack{|z'| < 1 \\ w(z'-z) \geq 0}} \frac{(1+wz)u'_i\left(\frac{T(w+z)}{w+z'}, z'\right) + x\,(z'-z)}{1 + z'w}, \frac{z^2+1}{2} \right\}$$

$$= \min \left\{ \inf_{\substack{x \in [-1,1]}} \max_{w = \pm 1} \inf_{\substack{|z'| < 1 \\ w(z'-z) \geq 0}} \frac{(1+wz)u_i\left(z'\right) + x\,(z'-z)}{1 + z'w}, \frac{z^2+1}{2} \right\}.$$

Note that the rightmost side does not depend on $T$. If we define $u_{i+1}(z)$ by the rightmost side of the above equation, we have $u'_{i+1}(T, z) = u_{i+1}(z)$. The proof is completed. $\qquad\square$

The function that plays a central role in our minimax analysis is the function $u_k(z)$ given in Lemma 16. We call it the *normalized minimax regret function*. By Lemma 10, Corollary 13 and Lemma 16, we have an immediate corollary.

**Corollary 17** (Recursive relation of normalized minimax regret). *If $|z| \geq 1$, $u_k(z) = |z|$. If $|z| < 1$, the normalized minimax regret satisfies*

$$u_{k+1}(z) = \min \left\{ \inf_{x \in [-1,1]} \max_{w = \pm 1} \inf_{\substack{|z'| < 1 \\ w(z'-z) \geq 0}} \frac{(1+wz)u_k(z') + x(z'-z)}{1 + z'w}, \frac{z^2 + 1}{2} \right\}. \quad (6)$$

**Lemma 18** (Boundary condition for $u_2$). *If $|z| < 1$, we have*

$$u_2(z) = \inf_{x \in [-1,1]} \max_{w = \pm 1} \inf_{\substack{|z'| < 1 \\ w(z'-z) \geq 0}} \frac{(1+wz) + x(z'-z)}{1 + z'w} = \frac{z^2 + 1}{2}.$$

*Proof.* Plugging $k = 1$ and $u_1(z) = 1$ for $|z| < 1$ into (6) gives

$$u_2(z) = \min \left\{ \inf_{x \in [-1,1]} \max_{w = \pm 1} \inf_{\substack{|z'| < 1 \\ w(z'-z) \geq 0}} \frac{(1+wz) + x(z'-z)}{1 + z'w}, \frac{z^2 + 1}{2} \right\}.$$

We define the function $f(z') = \frac{(1+wz)+x(z'-z)}{1+z'w}$. Differentiating this function yields

$$\frac{df}{dz'} = -\frac{(w-x)(wz+1)}{(wz'+1)^2}.$$

Since $w = \pm 1$, $|x| \leq 1$ and $|z| < 1$, we have $wz + 1 > 0$ and that the sign of $\frac{df}{dz}$ is the same as $-\text{sign}(w)$, or 0 if $w = x$. Therefore, the function is non-decreasing if $w = -1$ and is non-increasing if $w = 1$. The innermost infimum is attained as $z' \to w$. As a result, we deduce

$$\inf_{\substack{|z'| < 1 \\ w(z'-z) \geq 0}} \frac{(1+wz) + x(z'-z)}{1 + z'w} = \frac{1}{2}(w(x+z) - xz + 1),$$

and

$$\max_{w = \pm 1} \inf_{\substack{|z'| < 1 \\ w(z'-z) \geq 0}} \frac{(1+wz) + x(z'-z)}{1 + z'w} = \frac{1}{2} \max_{w = \pm 1} (w(x+z) - xz + 1)$$

$$= \frac{1}{2} \max\{x(1-z) + z + 1, -x(z+1) - z + 1\}.$$

The outermost infimum is attained when $x(1-z) + z + 1 = -x(z+1) - z + 1$, or equivalently, at $x = -z$. Therefore, we have

$$\inf_{x \in [-1,1]} \max_{w = \pm 1} \inf_{\substack{|z'| < 1 \\ w(z'-z) > 0}} \frac{(1+wz) + x(z'-z)}{1 + z'w} = \frac{1}{2}[x(1-z) + z + 1]_{x=-z} = \frac{z^2 + 1}{2}.$$

Therefore $u_2(z)$ also equals $\frac{z^2+1}{2}$. $\qquad\square$

**Lemma 19** (Monotonicity in $k$). *The sequence of functions $u_k(z)$ is non-increasing pointwise on $(-1, 1)$, i.e., $u_{k+1}(z) \leq u_k(z)$ for $|z| < 1$.*

*Proof.* By the definition of $r_k$ in (2), we see that a player's strategy with $k$ switches can be viewed as a strategy with $k + 1$ switches. Therefore, we have $r_{k+1}(T, z) \leq r_k(T, z)$ and therefore $u_{k+1}(z) = \frac{1}{T}r_{k+1}(T, z) \leq \frac{1}{T}r_k(T, z) = u_k(z)$. $\qquad\square$

Combining Lemma 18 and Lemma 19 implies the following corollary immediately.

**Corollary 20.** *For all $k \geq 2$ and $|z| < 1$, $u_k(z) \leq \frac{z^2+1}{2}$.*

Lemma 21 improves the recursive relation in Corollary 17 by removing the operation of taking the minimum with $\frac{z^2+1}{2}$. In fact, Lemma 21 and Corollary 17 are mathematically equivalent since we will show that the first term in the minimum operator in (6) is always less than or equal to the second term $\frac{z^2+1}{2}$.

**Lemma 21** (Improved recursive relation of $u_k$). *For all $k \geq 1$ and $|z| < 1$, $u_k(z)$ obeys the recursive relation*

$$u_{k+1}(z) = \inf_{x \in [-1,1]} \max_{w=\pm 1} \inf_{\substack{|z'|<1 \\ w(z'-z) \geq 0}} \frac{(1+wz)u_k(z') + x(z'-z)}{1+z'w}. \tag{7}$$

*Proof.* If $k = 1$, the desired equation holds due to Lemma 18. If $k \geq 2$, Corollary 20 shows $u_k(z) \leq \frac{z^2+1}{2}$. If we take $z' = z$, we have

$$\inf_{\substack{|z'|<1 \\ w(z'-z) \geq 0}} \frac{(1+wz)u_k(z') + x(z'-z)}{1+z'w} \leq \inf_{\substack{|z'|<1 \\ w(z'-z) \geq 0}} \frac{(1+wz)\frac{z^2+1}{2} + x(z'-z)}{1+z'w}$$

$$\leq \left[ \frac{(1+wz)\frac{z^2+1}{2} + x(z'-z)}{1+z'w} \right]_{z'=z}$$

$$= \frac{z^2+1}{2}.$$

By Corollary 17, we deduce

$$u_{k+1}(z) = \min \left\{ \inf_{x \in [-1,1]} \max_{w=\pm 1} \inf_{\substack{|z'|<1 \\ w(z'-z) \geq 0}} \frac{(1+wz)u_k(z') + x(z'-z)}{1+z'w}, \frac{z^2+1}{2} \right\}$$

$$= \inf_{x \in [-1,1]} \max_{w=\pm 1} \inf_{\substack{|z'|<1 \\ w(z'-z) \geq 0}} \frac{(1+wz)u_k(z') + x(z'-z)}{1+z'w}.$$

$\square$

### C.5 Fugal Operator and Quadratic Lower Bound

The recursive relation in Lemma 21 relates two consecutive $u_k$'s. In light of this recursive relation, we define the fugal operator that sends $u_k$ to $u_{k+1}$.

**Definition 2** (Fugal operator). Let $C[-1, 1]$ denote the space of continuous functions on $[-1, 1]$. The *fugal operator* $\mathcal{T} : C[-1, 1] \to C[-1, 1]$ is defined by

$$(\mathcal{T}f)(z) \triangleq \inf_{x \in [-1,1]} \max_{w=\pm 1} \inf_{\substack{|z'|<1 \\ w(z'-z) \geq 0}} \frac{(1+wz)f(z') + x(z'-z)}{1+z'w},$$

where $f \in C[-1, 1]$.

**Remark 2.** Using this notation, Lemma 21 can be re-written in a more compact way

$$u_{k+1} = \mathcal{T}u_k.$$

**Remark 3** (Monotonicity of fugal operator). If $f, g \in C[-1, 1]$ satisfy $f(z) \leq g(z)$ for all $z \in [-1, 1]$, we have the following inequality $\frac{(1+wz)f(z')+x(z'-z)}{1+z'w} \geq \frac{(1+wz)g(z')+x(z'-z)}{1+z'w}$. This is because $1 + wz \geq 0$ holds for any $w = \pm 1$ and $|z| \leq 1$, and $1 + z'w > 0$ holds for any $w = \pm 1$ and $|z'| < 1$. As a result, we have $(\mathcal{T}f)(z) \geq (\mathcal{T}g)(z)$ for all $z \in [-1, 1]$.

Before deriving a lower bound for $u_k$, we study the action of the fugal operator on *quadratic lower bound functions*.

**Definition 3** (Quadratic lower bound functions). The quadratic lower bound functions $a_k(z)$ on $[-1, 1]$ are defined by by $a_1(z) = 1$ and for $i \geq 2$

$$a_i(z) = \begin{cases} \frac{\sqrt{i/2}z^2 + \sqrt{2/i}}{2}, & |z| < \sqrt{2/i}, \\ |z|, & |z| \geq \sqrt{2/i}. \end{cases}$$

**Remark 4** (Continuity). If $z = \pm\sqrt{2/i}$, the expression $\frac{\sqrt{i/2}z^2 + \sqrt{2/i}}{2} = \sqrt{2/i} = |z|$. The quadratic lower bound function $a_i$ is continuous on $[-1, 1]$.

**Remark 5.** If $i = 1, 2$, the quadratic lower bound functions agree with the normalized minimax regret functions, *i.e.*, $a_1(z) = u_1(z) = 1$ and $a_2(z) = u_2(z) = \frac{z^2+1}{2}$.

We will show later in Lemma 28 that the quadratic lower bound functions provide indeed a lower bound for $u_k$'s, *i.e.*, $a_i(z) \leq u_i(z)$. This result will be proved in two steps. The first step is to obtain the closed-form expression of $\mathcal{T}a_i$ (Proposition 22) and the second step is to show that the fugal operator interlaces $a_i$, in other words, $\mathcal{T}a_i \geq a_{i+1}$ (Lemma 27). Then we can argue that $u_{i+1} = \mathcal{T}u_i \geq \mathcal{T}a_i \geq a_{i+1}$, provided that $u_i \geq a_i$, where the first inequality is due to the monotonicity of the fugal operator and the second inequality is because the fugal operator interlaces $a_i$. Therefore $a_i \leq u_i$ for all $i$ can be obtained by induction.

**Proposition 22** (Fugal operator on quadratic lower bound functions). *If $i \geq 2$, it holds that*

$$(\mathcal{T}a_i)(z) = \begin{cases} \sqrt{\frac{i}{2}}\left[z^2 - 1 + \sqrt{1 + \frac{2}{i} - z^2}\right], & |z| \leq \sqrt{2/i}, \\ |z|, & |z| > \sqrt{2/i}. \end{cases}$$

Before presenting the proof of Proposition 22, we need several lemmas.

**Lemma 23.** *If $i \geq 2$ and we define*

$$z_+ \triangleq \sqrt{1 + \frac{2}{i} - 2\sqrt{\frac{2}{i}}x} - 1$$

$$z_- \triangleq 1 - \sqrt{1 + \frac{2}{i} + 2\sqrt{\frac{2}{i}}x}$$

$$g_+(x, z) \triangleq x + (1 + z)\left[\frac{a_i(z') - x}{1 + z'}\right]_{z'=\max\{z, z_+\}}$$

$$g_-(x, z) \triangleq -x + (1 - z)\left[\frac{a_i(z') + x}{1 - z'}\right]_{z'=\min\{z, z_-\}},$$

*the following equation holds*

$$(\mathcal{T}a_i)(z) = \inf_{x \in [-1, 1]} \max\{g_+(x, z), g_-(x, z)\}. \tag{8}$$

*Proof.* Recalling the definition of the fugal operator gives

$$(\mathcal{T}a_i)(z)$$

$$= \inf_{x \in [-1, 1]} \max_{w=\pm 1} \inf_{\substack{|z'| < 1 \\ w(z'-z) \geq 0}} \frac{(1 + wz)a_i(z') + x(z' - z)}{1 + z'w}$$

$$= \inf_{x \in [-1, 1]} \max\left\{\inf_{z':z \leq z' < 1} \frac{(1 + z)a_i(z') + x(z' - z)}{1 + z'}, \inf_{-1 < z' \leq z} \frac{(1 - z)a_i(z') + x(z' - z)}{1 - z'}\right\}$$

We observe that if $|z'| < 1$, the following equations hold

$$\frac{(1 + z)a_i(z') + x(z' - z)}{1 + z'} = \frac{1 + z}{1 + z'}(a_i(z') - x) + x,$$

$$\frac{(1 - z)a_i(z') + x(z' - z)}{1 - z'} = \frac{1 - z}{1 - z'}(a_i(z') + x) - x.$$

Therefore, we simplify the innermost infima

$$\inf_{z':z\leq z'<1} \frac{(1+z)a_i(z')+x(z'-z)}{1+z'} = x + (1+z) \inf_{z':z\leq z'<1} \frac{a_i(z')-x}{1+z'},$$

$$\inf_{z':-1<z'\leq z} \frac{(1-z)a_i(z')+x(z'-z)}{1-z'} = -x + (1-z) \inf_{z':-1<z'\leq z} \frac{a_i(z')+x}{1-z'}.$$

We define two functions $f_+(z') = \frac{a_i(z')-x}{1+z'}$ and $f_-(z') = \frac{a_i(z')+x}{1-z'}$.

First, we assume $|z'| < \sqrt{2/i}$. Differentiating $f_+$ gives

$$\frac{df_+}{dz'} = \frac{\sqrt{2i}z'(z'+2) - 2\sqrt{\frac{2}{i}} + 4x}{4(z'+1)^2}.$$

Setting $\frac{df_+}{dz'} > 0$ yields

$$(z'+1)^2 > 1 + \frac{2}{i} - 2\sqrt{\frac{2}{i}}x.$$

The fact that $|x| \leq 1$ implies $1 + \frac{2}{i} - 2\sqrt{\frac{2}{i}}x \geq 1 + \frac{2}{i} - 2\sqrt{\frac{2}{i}} = \left(\sqrt{\frac{2}{i}}-1\right)^2 \geq 0$. Therefore $\frac{df_+}{dz'} > 0$ is equivalent to

$$z'+1 = |z'+1| > \sqrt{1 + \frac{2}{i} - 2\sqrt{\frac{2}{i}}x}.$$

In other words, $\frac{df_+}{dz'} > 0$ if and only if $z' > z_+ \triangleq \sqrt{1 + \frac{2}{i} - 2\sqrt{\frac{2}{i}}x} - 1$. Our assumption $i \geq 2$ implies

$$\sqrt{1 + \frac{2}{i} - 2\sqrt{\frac{2}{i}}x} \geq \sqrt{1 + \frac{2}{i} - 2\sqrt{\frac{2}{i}}} = \left|1 - \sqrt{\frac{2}{i}}\right| = 1 - \sqrt{\frac{2}{i}}.$$

As a result, $z_+ \geq -\sqrt{\frac{2}{i}}$. Since

$$\sqrt{1 + \frac{2}{i} - 2\sqrt{\frac{2}{i}}x} \leq \sqrt{1 + \frac{2}{i} + 2\sqrt{\frac{2}{i}}} = \left|\sqrt{\frac{2}{i}} + 1\right| = 1 + \sqrt{\frac{2}{i}},$$

we obtain the upper bound $z_+ \leq \sqrt{2/i}$, where the second inequality uses the assumption $i \geq 2$. Thus we are certain that $|z_+| \leq \sqrt{2/i}$.

If $z' \geq \sqrt{2/i}$, the function $f_+$ becomes $f_+(z') = \frac{z'-x}{1+z'} = 1 - \frac{1+x}{1+z'}$, which is non-decreasing in $z'$. On the other hand, if $z' \leq -\sqrt{2/i}$, the function $f_+$ becomes $f_+(z') = \frac{-z'-x}{1+z'} = -1 + \frac{1-x}{1+z'}$, which is non-increasing in $z'$. It follows that $f_+$ is non-increasing on $(-1, z_+)$ and non-decreasing on $(z_+, 1)$. Therefore we can solve the infimum

$$\inf_{z':z\leq z'<1} \frac{a_i(z')-x}{1+z'} = \left[\frac{a_i(z')-x}{1+z'}\right]_{z'=\max\{z,z_+\}}.$$

If $|z'| < \sqrt{2/i}$, the derivative of $f_-$ with respect to $z'$ is

$$\frac{df_-}{dz'} = \frac{4\sqrt{i}x - \sqrt{2}i(z'-2)z' + 2\sqrt{2}}{4\sqrt{i}(z'-1)^2}.$$

Setting the derivative greater than 0 yields

$$(z'-1)^2 < 1 + \frac{2}{i} + 2\sqrt{\frac{2}{i}}x.$$

The right-hand side is at least $1 + \frac{2}{i} - 2\sqrt{\frac{2}{i}} = (1 - \sqrt{2/i})^2 \geq 0$. Since the right-hand side is non-negative and $z < 1$, we have

$$z' > z_- \triangleq 1 - \sqrt{1 + \frac{2}{i} + 2\sqrt{\frac{2}{i}}x}\,.$$

If $z' \geq \sqrt{2/i}$, the function $f_-$ equals $\frac{z'+x}{1-z'} = -1 + \frac{x+1}{1-z'}$, which is non-decreasing in $z'$. On the other hand, if $z' \leq -\sqrt{2/i}$, the function $f_-$ equals $\frac{-z'+x}{1-z'} = 1 + \frac{x-1}{1-z'}$, which is non-increasing in $z'$. It follows that $f_-$ is non-increasing on $(-1, z_-)$ and non-decreasing on $(z_-, 1)$. Thus we solve the other infimum

$$\inf_{z':-1<z'\leq z} \frac{a_i(z') + x}{1 - z'} = \left[\frac{a_i(z') + x}{1 - z'}\right]_{z'=\min\{z,z_-\}}\,.$$

The equation (8) is thereby obtained by combining our results regarding the two infima. $\qquad\square$

**Lemma 24.** *If $z_+$ and $z_-$ are as defined in Lemma 23, we have $z_+ \geq z_-$.*

*Proof.* We compute the difference of $z_+$ and $z_-$

$$z_+ - z_- = \sqrt{1 + \frac{2}{i} + 2\sqrt{\frac{2}{i}}x} + \sqrt{1 + \frac{2}{i} - 2\sqrt{\frac{2}{i}}x} - 2\,.$$

To show that $z_+ - z_- \geq 0$, it is sufficient to show that

$$\left(\sqrt{1 + \frac{2}{i} + 2\sqrt{\frac{2}{i}}x} + \sqrt{1 + \frac{2}{i} - 2\sqrt{\frac{2}{i}}x}\right)^2 \geq 4\,.$$

The left-hand side equals

$$2\left(1 + \frac{2}{i}\right) + 2\sqrt{\left(1 + \frac{2}{i}\right)^2 - 4\cdot\frac{2}{i}x^2} \geq 2\left(1 + \frac{2}{i}\right) + 2\sqrt{\left(1 + \frac{2}{i}\right)^2 - 4\cdot\frac{2}{i}}$$

$$= 2\left(1 + \frac{2}{i} + \left|1 - \frac{2}{i}\right|\right) = 4\,,$$

where the last inequality is because $i \geq 2$ and thus $1 - \frac{2}{i} \geq 0$. Therefore we establish $z_+ \geq z_-$. $\quad\square$

**Lemma 25.** *Given $|z| \leq 1$, the function $h_z(x) \triangleq g_+(x, z) - g_-(x, z)$ has a unique zero $x = x_0(z)$ on $[-1, 1]$ and it satisfies*

$$(\mathcal{T}a_i)(z) = \inf_{x\in[-1,1]} \max\{g_+(x, z), g_-(x, z)\} = g_+(x_0(z), z) = g_-(x_0(z), z)\,.$$

*Proof.* Recall that $z_+$ and $z_-$ are functions of $x$ but do not rely on $z$. They obey an additional relation

$$z_+(x) + z_-(-x) = 0\,.$$

Their inverses are

$$z_+^{-1}(z) = \frac{2 - iz^2 - 2iz}{2\sqrt{2i}}, \quad z_-^{-1}(z) = \frac{iz^2 - 2iz - 2}{2\sqrt{2i}}\,,$$

respectively. Both inverse functions are strictly decreasing. Using the relation $z_+(x) + z_-(-x) = 0$, since $\max\{-z, z_+(-x)\} = \max\{-z, -z_-(x)\} = -\min\{z, z_-(x)\}$ and $a_i$ is an even function, we have

$$g_+(-x, -z) = -x + (1 - z)\left[\frac{a_i(z') + x}{1 + z'}\right]_{z'=\max\{-z,z_+(-x)\}}$$

$$= -x + (1 - z)\left[\frac{a_i(z') + x}{1 + z'}\right]_{z'=-\min\{z,z_-(x)\}} \qquad (9)$$

$$= -x + (1 - z)\left[\frac{a_i(-z') + x}{1 - z'}\right]_{z'=\min\{z,z_-(x)\}}$$

$$= g_-(x, z)\,.$$

If $z \geq z_+$, we have $z' = z$ and $g_+(x, z) = a_i(z)$. In this case, $g_+$ is a constant function with respect to $x$. If $z < z_+$, we have $z' = z_+$ and $g_+(x, z) = x + (1+z)\frac{a_i(z_+)-x}{1+z_+}$. Since $|z_+| \leq \sqrt{2/i}$, we have $a_i(z_+) = \frac{\sqrt{i/2}z_+^2 + \sqrt{2/i}}{2}$ and

$$g_+(x, z) = x + (1+z)\frac{(\sqrt{i/2}z_+^2 + \sqrt{2/i})/2 - x}{1+z_+}.$$

Differentiating $g_+$ yields

$$\frac{\partial g_+}{\partial x} = 1 - \frac{i(z+1)}{\sqrt{i\left(-2\sqrt{2i}x + i + 2\right)}}.$$

Since $z < z_+ = \sqrt{1 + \frac{2}{i} - 2\sqrt{\frac{2}{i}}x} - 1$, we have

$$z + 1 < \sqrt{1 + \frac{2}{i} - 2\sqrt{\frac{2}{i}}x},$$

which, in turn, implies

$$\frac{\partial g_+}{\partial x} > 1 - \frac{i\sqrt{1 + \frac{2}{i} - 2\sqrt{\frac{2}{i}}x}}{\sqrt{i\left(-2\sqrt{2i}x + i + 2\right)}} = 0.$$

Therefore, $g_+(x, z)$ is strictly increasing in $x$ if $z < z_+(x)$ (*i.e.*, $x < z_+^{-1}(z)$) and is constant with respect to $x$ if $z \geq z_+(x)$ (*i.e.*, $x \geq z_+^{-1}(z)$). Furthermore, we verify that $g_+(-1, z) = z$ and $g_+(1, z) = a_i(z)$.

In light of the relation (9), we derive the property of $g_-$. The function $g_-(x, z)$ is strictly decreasing if $-z < z_+(-x)$, or equivalently, $z > z_-(x)$ (*i.e.*, $x > z_-^{-1}(z)$). It stays at $a_i(z)$ if $z \leq z_-(x)$ (*i.e.*, $x \leq z_-^{-1}(z)$). Furthermore, we have $g_-(-1, z) = g_+(1, -z) = a_i(z)$ and $g_-(1, z) = g_+(-1, -z) = -z$.

Let $h_z(x) \triangleq g_+(x, z) - g_-(x, z)$ be the difference of these two functions. Since $g_+$ is non-decreasing in $x$ and $g_-$ is non-increasing in $x$, the function $h_z$ is non-decreasing in $x$. Then we check the value of $h_z$ at $x = -1$ and $x = 1$. We have $h_z(-1) = g_+(-1, z) - g_-(-1, z) = z - a_i(z)$ and $h_z(1) = g_+(1, z) - g_-(1, z) = a_i(z) + z$. Their product is $h_z(-1)h_z(1) = z^2 - a_i^2(z)$, which is non-positive because $|z| \leq a_i(z)$. The continuity of $h_z$ implies the existence of a zero on $[-1, 1]$. Next, we will show the uniqueness of the zero. Since $g_+$ is strictly increasing with respect to $x$ at the initial stage when $x < z_+^{-1}(z)$ and stays constant when $x \geq z_+^{-1}(z)$, and $g_-$ is constant with respect to $x$ at the initial stage when $x \leq z_-^{-1}(z)$ and strictly decreases when $x > z_-^{-1}(z)$, the only possibility of having more than one zero is that $z_-^{-1}(z) > z_+^{-1}(z)$ and that the set $R = [z_+^{-1}(z), z_-^{-1}(z)] \cap [-1, 1]$ contains more than one point. The inequality $z_-^{-1}(z) > z_+^{-1}(z)$ is equivalent to $|z| > \sqrt{2/i}$. A necessary condition for the set $R$ containing more than one point is that both $z_-^{-1}(z) > -1$ and $z_+^{-1}(z) < 1$ holds. If $i = 2$, $|z| > \sqrt{2/i} = 1$ will never happen. If $i > 2$, the expression $z_-^{-1}(z) > -1$ is equivalent to $-1 \leq z < \sqrt{\frac{2}{i}}$ while the expression $z_+^{-1}(z) < 1$ is equivalent to $-\sqrt{\frac{2}{i}} < z \leq 1$. However, the three inequalities $|z| > \sqrt{2/i}$, $-1 \leq z < \sqrt{\frac{2}{i}}$, and $-\sqrt{\frac{2}{i}} < z \leq 1$ cannot be satisfied simultaneously. Therefore, we show that $h_z(x)$ has a unique zero on $[-1, 1]$. Let $x_0(z)$ denote the unique zero, which is a function of $z$. By its definition, the two functions $g_+$ and $g_-$ are equal at $x = x_0$. Since $h_z$ is non-decreasing with respect to $x$ and $x_0$ is the unique zero, we know that $g_+(x) > g_-(x)$ if $x > x_0$ and $g_+(x) < g(x)$ if $x < x_0$. Therefore, by Lemma 23, $(\mathcal{T}a_i)(z)$ equals

$$(\mathcal{T}a_i)(z) = \inf_{x \in [-1, 1]} \max\{g_+(x, z), g_-(x, z)\} = g_+(x_0(z), z) = g_-(x_0(z), z).$$

$\square$

We are now ready to prove Proposition 22.

*Proof of Proposition 22.* In light of Lemma 25, we compute the closed-form expression of $\mathcal{T}a_i$ by verifying that

$$x_0(z) = \begin{cases} -\frac{z\sqrt{-iz^2+i+2}}{\sqrt{2}}, & |z| \le \sqrt{2/i}, \\ -\operatorname{sign}(z), & |z| > \sqrt{2/i}, \end{cases}$$

is the unique zero of $h_z(x)$. We consider two cases $|z| \le \sqrt{2/i}$ and $|z| > \sqrt{2/i}$.

**Case 1:** $|z| > \sqrt{2/i}$. Let us begin with the case where $|z| > \sqrt{2/i}$. In this case, $x_0(z) = -\operatorname{sign}(z)$ and it is indeed on $[-1, 1]$. We further divide this case into two sub-cases where $z > \sqrt{2/i}$ and $z < -\sqrt{2/i}$, respectively.

**Case 1.1:** $z > \sqrt{2/i}$. If $z > \sqrt{2/i}$, and since $|z_+| \le \sqrt{2/i}$, we know that $z > z_+$ and $\max\{z, z_+\} = z$. In this sub-case, we have $x_0 = -1$ and

$$g_+(x_0, z) = -1 + (1+z)\frac{a_i(z)+1}{1+z} = a_i(z) = z.$$

Since $|z_-| \le \sqrt{2/i}$ and $z > z_-$, we have $\min\{z, z_-\} = z_-$. Therefore, we can compute

$$z_- = z_-(-1) = 1 - \sqrt{1 + \frac{2}{i} - 2\sqrt{\frac{2}{i}}} = \sqrt{\frac{2}{i}}$$

and

$$g_-(x_0, z) = 1 + (1-z)\frac{a_i(z_-)-1}{1-z_-} = 1 + (1-z)\frac{\sqrt{2/i}-1}{1-\sqrt{2/i}} = z.$$

**Case 1.2:** $z < -\sqrt{2/i}$. In the second sub-case, we assume that $z < -\sqrt{2/i}$. In this sub-case, we have $x_0 = 1$, $\max\{z, z_+\} = z_+ = -\sqrt{2/i}$, and $\min\{z, z_-\} = z$. The function $g_+(x_0, z)$ equals

$$g_+(x_0, z) = 1 + (1+z)\frac{a_i(-\sqrt{2/i})-1}{1-\sqrt{2/i}} = -z,$$

where the function $g_-(x_0, z)$ equals

$$g_-(x_0, z) = -1 + (1-z)\frac{a_i(z)+1}{1-z} = a_i(z) = -z.$$

Therefore, $x_0 = -\operatorname{sign}(z)$ is indeed the root when $|z| > \sqrt{2/i}$. Combining these two sub-cases, we deduce that if $|z| > \sqrt{2/i}$,

$$(\mathcal{T}a_i)(z) = |z|. \tag{10}$$

**Case 2:** $|z| \le \sqrt{2/i}$. The case that needs more work is $|z| \le \sqrt{2/i}$. In this case, the root function is $x_0(z) = -\frac{z\sqrt{-iz^2+i+2}}{\sqrt{2}}$. First, let us check that $x_0(z)$ resides on $[-1, 1]$. Since $|z| \le \sqrt{2/i}$, it holds that $(z^2-1)(iz^2-2) \ge 0$. Expanding it and re-arranging the terms yields $z^2(i+2-iz^2) \le 2$ and therefore $|x_0(z)| \le 1$.

We claim $z_-^{-1}(z) \le x_0(z) \le z_+^{-1}(z)$. Notice the following factorization

$$x_0(z) - z_-^{-1}(z) = \frac{\left(\sqrt{-iz^2+i+2} - \sqrt{i}\right)\left(\sqrt{-iz^2+i+2} - 2\sqrt{i}z + \sqrt{i}\right)}{2\sqrt{2i}}.$$

The first term $\sqrt{-iz^2+i+2} - \sqrt{i}$ is a decreasing function with respect to $z^2$. Since $z^2 \le 2/i$, the first term is non-negative. Let $s(z) \triangleq \sqrt{-iz^2+i+2} - 2\sqrt{i}z + \sqrt{i}$ denote the second term. Its derivative is $s'(z) = -\frac{iz}{\sqrt{-iz^2+i+2}} - 2\sqrt{i}$. If $z \ge 0$, we see that $s'(z) < 0$. Since $i \ge 2$, it holds that $2/i \le 4(1+2/i)/5$. In light of the assumption $z^2 \le 2/i$, we get $z^2 \le 4(1+2/i)/5$. Re-arranging the terms gives $z^2 \le 4(1+2/i-z^2)$. If $z < 0$, taking the square root of both sides yields $-z \le 2\sqrt{1+2/i-z^2}$. Re-arranging the terms again proves that if $z < 0$, $s'(z) \le 0$. Since

$s(z)$ is a continuous function, we show that $s$ is a non-increasing function on $[-\sqrt{2/i}, \sqrt{2/i}]$ and that for any $z \in [-\sqrt{2/i}, \sqrt{2/i}]$, we have $s(z) \geq s(\sqrt{2/i}) = 2(\sqrt{i} - \sqrt{2}) \geq 0$. Thus we show that $x_0(z) \geq z_-^{-1}(z)$.

Next we need to show that $x_0(z) \leq z_+^{-1}(z)$. Notice the following factorization

$$z_+^{-1}(z) - x_0(z) = \frac{\left(\sqrt{-iz^2 + i + 2} - \sqrt{i}\right)\left(\sqrt{-iz^2 + i + 2} + 2\sqrt{i}z + \sqrt{i}\right)}{2\sqrt{2i}}.$$

We observe that $z_+^{-1}(z) - x_0(z) = x_0(-z) - z_-^{-1}(-z) \geq 0$ since $-z$ is also on $[-\sqrt{2/i}, \sqrt{2/i}]$. Therefore we conclude that for all $z \in [-\sqrt{2/i}, \sqrt{2/i}]$, the inequality $z_-^{-1}(z) \leq x_0(z) \leq z_+^{-1}(z)$ holds. This inequality implies

$$z_-(x_0(z)) \leq z \leq z_+(x_0(z)).$$

In what follows, we compute the exact values of $z_+(x_0(z))$ and $z_-(x_0(z))$. We first compute $z_+(x_0(z))$

$$z_+(x_0(z)) = \sqrt{2z\sqrt{-z^2 + 1 + \frac{2}{i}} + \frac{2}{i} + 1} - 1$$

$$= \left|\sqrt{-z^2 + 1 + \frac{2}{i}} + z\right| - 1$$

$$= \sqrt{-z^2 + 1 + \frac{2}{i}} + z - 1.$$

The last equality is because $\sqrt{-z^2 + 1 + \frac{2}{i}} + z \geq 0$. To see this, we define $s_1(z) \triangleq \sqrt{-z^2 + 1 + \frac{2}{i}} + z$. Its second derivative is $s_1''(z) = \frac{i+2}{\sqrt{\frac{2}{i} - z^2 + 1}(i(z^2 - 1) - 2)} \leq 0$. Therefore, for any $z \in [-\sqrt{2/i}, \sqrt{2/i}]$, its derivative satisfies $s_1'(z) \geq s_1'(\sqrt{2/i}) = 1 - \sqrt{2/i} \geq 0$. As a result, for any $z \in [-\sqrt{2/i}, \sqrt{2/i}]$, $s_1(z) \geq s_1(-\sqrt{2/i}) = 1 - \sqrt{2/i} \geq 0$. On the other hand, we compute $z_-(x_0(z))$

$$z_-(x_0(z)) = 1 - \sqrt{-2z\sqrt{-z^2 + 1 + \frac{2}{i}} + \frac{2}{i} + 1}$$

$$= 1 - \left|\sqrt{-z^2 + 1 + \frac{2}{i}} - z\right|$$

$$= -\sqrt{\frac{2}{i} - z^2 + 1} + z + 1.$$

The last inequality is because $\sqrt{-z^2 + 1 + \frac{2}{i}} - z = s_1(-z) \geq 0$.

Now, let us compute $g_+(x_0(z), z)$ and $g_-(x_0(z), z)$. Since $\max\{z, z_+(x_0)\} = z_+(x_0)$ and $a_i(z_+) = \frac{\sqrt{i/2z_+^2} + \sqrt{2/i}}{2}$ (this is because $|z_+| \leq \sqrt{2/i}$), plugging $z_+(x_0(z))$ into the definition of $g_+(x_0(z), z)$ yields

$$g_+(x_0(z), z) = \frac{-\sqrt{i(-iz^2 + i + 2)} + (z+1)\left(z\sqrt{i(-iz^2 + i + 2)} + i(z-1)^2\right) + 2}{\sqrt{2}\left(\sqrt{-iz^2 + i + 2} + \sqrt{i}z\right)}. \quad (11)$$

Let $A = \sqrt{-iz^2 + i + 2}$. Solving $z$ out of this expression, we get $z = \pm\frac{\sqrt{-A^2 + i + 2}}{\sqrt{i}}$. Plugging it into (11), we obtain

$$g_+(x_0(z), z) = \frac{A\sqrt{i} - A^2 + 2}{\sqrt{2i}}.$$

Note that the result remains invariant no matter whether we plug in $z = \frac{\sqrt{-A^2+i+2}}{\sqrt{i}}$ or $z = -\frac{\sqrt{-A^2+i+2}}{\sqrt{i}}$. We plug in the definition of $A$ and express $g_+(x_0(z), z)$ in terms of $z$ again

$$g_+(x_0(z), z) = \sqrt{\frac{i}{2}} \left[ z^2 - 1 + \sqrt{1 + \frac{2}{i} - z^2} \right] .$$

Similarly, since $\min\{z, z_-(x_0)\} = z_-(x_0)$ and $a_i(z_-) = \frac{\sqrt{i/2}z_-^2 + \sqrt{2/i}}{2}$, plugging $z_-(x_0(z))$ into the definition of $g_-(x_0(z), z)$ yields

$$g_-(x_0(z), z) = \frac{(z-1)z\sqrt{i(-iz^2+i+2)} - \sqrt{i(-iz^2+i+2)} - i(z-1)(z+1)^2 + 2}{\sqrt{2}\left(\sqrt{-iz^2+i+2} - \sqrt{i}z\right)} . \quad (12)$$

Again plugging $z = \pm\frac{\sqrt{-A^2+i+2}}{\sqrt{i}}$ into (12) gives

$$g_-(x_0(z), z) = \frac{A\sqrt{i} - A^2 + 2}{\sqrt{2i}} ,$$

which equals $g_+(x_0(z), z)$. Therefore, we conclude that if $|z| \le \sqrt{2/i}$,

$$(\mathcal{T}a_i)(z) = \sqrt{\frac{i}{2}} \left[ z^2 - 1 + \sqrt{1 + \frac{2}{i} - z^2} \right] . \quad (13)$$

Combining (8), (10) and (13), we establish

$$(\mathcal{T}a_i)(z) = \begin{cases} \sqrt{\frac{i}{2}} \left[ z^2 - 1 + \sqrt{1 + \frac{2}{i} - z^2} \right], & |z| \le \sqrt{2/i}, \\ |z|, & |z| > \sqrt{2/i}. \end{cases}$$

$\square$

### C.6 Exact Values of Normalized Minimax Regret

Recall that $u_2(z) = a_2(z)$. Proposition 22 implies

$$u_3(z) = (\mathcal{T}u_2)(z) = (\mathcal{T}a_2)(z) = \begin{cases} z^2 - 1 + \sqrt{2 - z^2}, & |z| \le 1, \\ |z|, & |z| > 1. \end{cases}$$

Therefore we have $u_1(0) = 1$, $u_2(0) = \frac{1}{2}$, and $u_3(0) = \sqrt{2} - 1$. These exact values imply that the minimax regret of a $T$-round fugal game is exactly $T$, $\frac{T}{2}$, and $(\sqrt{2} - 1)T$ if the player is allowed to switch at most 0, 1, and 2 times, respectively. In Proposition 26, we compute the exact value of $u_4(0)$. The complicated form of $u_4(0)$ suggests that it is highly challenging to find a pattern for the general form of $u_i(0)$ and that we should consider lower bounds whose behavior under the action of the fugal operator is more amenable to analysis, as what we will discuss in Appendix C.7.

**Proposition 26.** *The value of $u_4(0)$ is given by*

$$u_4(0)$$
$$= \frac{1}{3}\sqrt[3]{45\sqrt{2} + 3\sqrt{3\left(502\sqrt{2} + 945\right)} + 145} - \frac{5}{3} - \frac{2\left(3\sqrt{2} + 1\right)}{3\sqrt[3]{45\sqrt{2} + 3\sqrt{3\left(502\sqrt{2} + 945\right)} + 145}}$$

$$\approx 0.362975 .$$

*Proof.* By the definition of the fugal operator, we have

$$u_4(0)$$
$$= (\mathcal{T}u_3)(0)$$
$$= \inf_{x\in[-1,1]} \max_{w=\pm 1} \inf_{\substack{|z'|<1 \\ wz'\geq 0}} \frac{u_3(z') + xz'}{1 + z'w}$$
$$= \inf_{x\in[-1,1]} \max_{w=\pm 1} \inf_{\substack{|z'|<1 \\ wz'\geq 0}} \frac{(z'^2 - 1 + \sqrt{2 - z'^2}) + xz'}{1 + z'w}$$
$$= \inf_{x\in[-1,1]} \max \left\{ \inf_{0\leq z'<1} \frac{(z'^2 - 1 + \sqrt{2 - z'^2}) + xz'}{1 + z'}, \inf_{-1<z'\leq 0} \frac{(z'^2 - 1 + \sqrt{2 - z'^2}) + xz'}{1 - z'} \right\}.$$

If we define $f(x, z') \triangleq \frac{(z'^2 - 1 + \sqrt{2 - z'^2}) + xz'}{1 + z'}$ and $g(x) \triangleq \inf_{0\leq z'<1} f(x, z')$, the value of $u_4(0)$ can be re-written as

$$u_4(0) = \inf_{x\in[-1,1]} \max \left\{ \inf_{0\leq z'<1} f(x, z'), \inf_{-1<z'\leq 0} f(-x, -z') \right\}$$
$$= \inf_{x\in[-1,1]} \max \left\{ \inf_{0\leq z'<1} f(x, z'), \inf_{0\leq z'<1} f(-x, z') \right\}$$
$$= \inf_{x\in[-1,1]} \max \left\{ g(x), g(-x) \right\}.$$

Note that $f(x, z)$ is non-decreasing with respect to $x$ provided that $z' \in [0, 1]$. Therefore, the function $g(x)$ is non-decreasing in $x$ and the inequality $g(x) \geq g(-x)$ is equivalent to $x \geq 0$. As a result, we deduce that

$$u_4(0) = \inf_{x\in[-1,1]} \max \left\{ g(x), g(-x) \right\} = g(0) = \inf_{0\leq z'<1} f(0, z') = \inf_{0\leq z'<1} \frac{z'^2 - 1 + \sqrt{2 - z'^2}}{1 + z'}.$$

The derivative of $f(0, z')$ with respect to $z'$ is $\frac{\partial f(0, z')}{\partial z'} = -\frac{z' + 2}{(z' + 1)^2 \sqrt{2 - z'^2}} + 1$. Setting this derivative greater than or equal to 0 yields a sextic polynomial $p(z') \triangleq -z'^6 - 4z'^5 - 4z'^4 + 4z'^3 + 10z'^2 + 4z' - 2 \geq 0$. By Descartes' rule of signs, this polynomial has two sign differences and thereby has two or zero positive roots. Since $p(0) = -2$, $p(1) = 7$ and $p(2) = -178$, we deduce that there is exactly one root in $(0, 1)$ and $(1, 2)$ respectively. Let $z_0$ denote the unique root of $p(z')$ in $(0, 1)$. The function $f(0, z')$ is decreasing on $[0, z_0]$ and increasing on $[z_0, 1]$. Thus the desired infimum $\inf_{0\leq z'<1} f(0, z')$ is attained at $f(0, z_0)$.

We notice that $p(z')$ can be factorized in $\mathbb{Q}(\sqrt{2})$ as below

$$p(z') = \left( z'^3 + 2z'^2 - \sqrt{2}z' - \sqrt{2} - 2 \right) \left( z'^3 + 2z'^2 + \sqrt{2}z' + \sqrt{2} - 2 \right).$$

Solving the two cubic polynomials with Cardano formula, we obtain the unique root in $(0, 1)$

$$z_0 = \frac{1}{6} \left( -2 \left( 3\sqrt{2} - 4 \right) \sqrt[3]{\frac{2}{-9\sqrt{2} + 3\sqrt{6 \left( 2\sqrt{2} + 9 \right)} + 38}} \right.$$
$$\left. + 2^{2/3} \sqrt[3]{-9\sqrt{2} + 3\sqrt{6 \left( 2\sqrt{2} + 9 \right)} + 38} - 4 \right)$$
$$\approx 0.283975.$$

Plugging $z' = z_0$ into $f(0, z')$ yields the desired expression for $u_4(0)$. $\qquad \square$

## C.7 Interlacing Quadratic Lower Bound Functions

**Lemma 27** (Fugal operator interlaces quadratic lower bound functions). *For $i \geq 1$ and $z \in [-1, 1]$, $a_{i+1}(z) \leq (\mathcal{T}a_i)(z)$.*

*Proof.* If $i = 1$, recall that $a_1(z) = 1$ and $a_2(z) = u_2(z) = \frac{z^2+1}{2}$. Lemma 18 implies $a_2(z) = (\mathcal{T}a_1)(z)$ and therefore the promised inequality holds. In the sequel, we assume that $i \geq 2$. In Proposition 22, we show that for $i \geq 2$,

$$(\mathcal{T}a_i)(z) = \begin{cases} \sqrt{\frac{i}{2}}\left[z^2 - 1 + \sqrt{1 + \frac{2}{i} - z^2}\right], & |z| \leq \sqrt{2/i}, \\ |z|, & |z| > \sqrt{2/i}. \end{cases}$$

Recall the definition of $a_{i+1}$

$$a_{i+1}(z) = \begin{cases} \frac{\sqrt{\frac{i+1}{2}}z^2 + \sqrt{\frac{2}{i+1}}}{2}, & |z| < \sqrt{\frac{2}{i+1}}, \\ |z|, & |z| \geq \sqrt{\frac{2}{i+1}}. \end{cases}$$

If $|z| > \sqrt{\frac{2}{i}}$, we have $(\mathcal{T}a_i)(z) = a_{i+1}(z)$. If $\sqrt{\frac{2}{i+1}} \leq |z| \leq \sqrt{\frac{2}{i}}$, we need to show that

$$(\mathcal{T}a_i)(z) - |z| = \sqrt{\frac{i}{2}}\left[z^2 - 1 + \sqrt{1 + \frac{2}{i} - z^2}\right] - |z| \geq 0.$$

Note that in this case, $l(z) \triangleq (\mathcal{T}a_i)(z) - |z|$ is an even function. Therefore it suffices to show the inequality for $\sqrt{\frac{2}{i+1}} \leq z \leq \sqrt{\frac{2}{i}}$. For any $y \in [0, 1]$ and $i \geq 2$, the following inequality holds

$$\frac{1}{\sqrt{\frac{2}{i}(1 - y^2) + 1}} + \frac{1}{y} - 2 \leq \lim_{i \to \infty}\left(\frac{1}{\sqrt{\frac{2}{i}(1 - y^2) + 1}} + \frac{1}{y} - 2\right) = \frac{1}{y} - 1 \geq 0.$$

Since $\sqrt{\frac{i}{2}}z \in [0, 1]$, setting $y = \sqrt{\frac{i}{2}}z$ in the above inequality gives

$$\frac{1}{\sqrt{\frac{2}{i} - z^2 + 1}} + \frac{\sqrt{2}}{\sqrt{i}z} - 2 \geq 0.$$

Re-arranging the terms, we get

$$\frac{dl}{dz} = -1 + \sqrt{\frac{2}{i}}z\left(2 - \frac{1}{\sqrt{\frac{2}{i} - z^2 + 1}}\right) \leq 0.$$

This implies that $l(z)$ is non-increasing if $z \leq \sqrt{\frac{2}{i}}$. Therefore, for any $\sqrt{\frac{2}{i+1}} \leq z \leq \sqrt{\frac{2}{i}}$, we have $l(z) \geq l(\sqrt{\frac{2}{i}}) = 0$.

If $|z| \leq \sqrt{\frac{2}{i+1}}$, we need to show that

$$(\mathcal{T}a_i)(z) - a_{i+1}(z) = \sqrt{\frac{i}{2}}\left[z^2 - 1 + \sqrt{1 + \frac{2}{i} - z^2}\right] - \frac{\sqrt{\frac{i+1}{2}}z^2 + \sqrt{\frac{2}{i+1}}}{2} \geq 0.$$

Since in this case the function $(\mathcal{T}a_i)(z) - a_{i+1}(z)$ is an even function with respect to $z$, we assume that $0 \leq z \leq \sqrt{\frac{2}{i+1}}$. Since $(\mathcal{T}a_i)(z) - a_{i+1}(z)$ is a concave function with respect to $z^2$ (note that $\sqrt{1 + \frac{2}{i} - z^2}$ is concave with respect to $z^2$ and that the remaining terms are linear in $z^2$), it is sufficient to check its non-negativity when $z^2 = 0$ and $z^2 = \frac{2}{i+1}$ (*i.e.*, when $z = 0$ and $z = \sqrt{\frac{2}{i+1}}$). Recall that we have shown that $(\mathcal{T}a_i)(z) - a_{i+1}(z) \geq 0$ holds for any $\sqrt{\frac{2}{i+1}} \leq z \leq \sqrt{\frac{2}{i}}$. It remains to check the non-negativity of $(\mathcal{T}a_i)(0) - a_{i+1}(0)$. We have $(\mathcal{T}a_i)(0) - a_{i+1}(0) = \frac{1}{\sqrt{2}}(-\sqrt{i} + \sqrt{i+2} - \frac{1}{\sqrt{i+1}})$. The concavity of the square root function implies $\sqrt{i+1} \geq \frac{\sqrt{i+2}+\sqrt{i}}{2} = \frac{1}{\sqrt{i+2}-\sqrt{i}}$. Thus we obtain $\sqrt{i+2} - \sqrt{i} \geq \frac{1}{\sqrt{i+1}}$ and the non-negativity of $(\mathcal{T}a_i)(0) - a_{i+1}(0)$. We conclude that $(\mathcal{T}a_i)(z) \geq a_{i+1}(z)$ in all three cases. $\qquad\square$

Lemma 28 shows that the quadratic lower bound functions indeed provide a lower bound for $u_k(z)$.

**Lemma 28** (Quadratic lower bound). *For all $k \geq 1$ and $|z| < 1$, it holds that $a_k(z) \leq u_k(z)$.*

*Proof.* If $k = 1$, the claim holds by recalling $u_k(z) = 1$ on $(-1, 1)$, as shown in Corollary 15. If $k = 2$, we have $a_2(z) = \frac{z^2+1}{2} \leq u_2(z)$ by Lemma 18, as promised. For $k > 2$, we will show the desired inequality by induction. Assume that $a_i(z) \leq u_i(z)$ for some $i \geq 2$. We will show that $a_{i+1}(z) \leq u_{i+1}(z)$. Since $a_i(z) \leq u_i(z)$, by Lemma 21, we deduce

$$u_{i+1}(z) = \inf_{x \in [-1,1]} \max_{w=\pm 1} \inf_{\substack{|z'|<1 \\ w(z'-z) \geq 0}} \frac{(1+wz)u_i(z') + x(z'-z)}{1+z'w}$$

$$\geq \inf_{x \in [-1,1]} \max_{w=\pm 1} \inf_{\substack{|z'|<1 \\ w(z'-z) \geq 0}} \frac{(1+wz)a_i(z') + x(z'-z)}{1+z'w} = (\mathcal{T}a_i)(z) \stackrel{(a)}{\geq} a_{i+1}(z),$$

where $(a)$ is due to Lemma 27. By induction, we know that $a_k(z) \leq u_k(z)$ holds for all $k \geq 1$ and $|z| < 1$.

$\square$

We are in a position to prove the minimax lower bound for the one-dimensional game.

*Proof of Proposition 9.* By Lemma 28, plugging $z = 0$ into $u_K(z) \geq a_K(z)$ shows that the normalized minimax regret without initial bias $u_K(0) \geq a_K(0) = \frac{1}{\sqrt{2K}}$. Recall that $u_K(0) = \frac{1}{T}r_K(T,0)$ for all $T > 0$, where $r_K(T,0)$ is the minimax regret with $T$ rounds, a maximum number of $K$ switches, and without initial bias. Therefore, we have $\frac{1}{T}r_K(T,0) \geq \frac{1}{\sqrt{2K}}$, which implies that $R_K(T,0) \geq \frac{T}{\sqrt{2K}}$ (because the minimax regret of switching-constrained online convex optimization is lower bounded by the minimax regret of a fugal game). $\square$

## C.8 Tightness of Lower Bound

In the following two propositions, we validate the one-dimensional lower bound of the previous section in two senses. First, in Proposition 4 we show that the constant in Proposition 9 cannot be increased for arbitrary $K$ and $T$. In particular, we demonstrate that when $K = 2$, the player has a simple strategy — playing 0 in the first half of the rounds, and an appropriately chosen constant in the second half — to guarantee regret no greater than $\lceil T/2 \rceil$.

**Proposition 4** (The constant in $\frac{T}{\sqrt{2K}}$ is unimprovable). *The constant $\frac{1}{\sqrt{2}}$ in the lower bound $\mathcal{R}(T, K) \geq \frac{T}{\sqrt{2K}}$ cannot be increased.*

*Proof.* We will show that the lower bound is tight when $K = 2$ by proving the upper bound $\mathcal{R}_{\text{OCO}}(B^1, B^{*1}, 2, T) \leq \lceil T/2 \rceil$. Recall that if $K = 2$, the lower bound $\frac{T}{\sqrt{2K}}$ is $T/2$. To prove the upper bound, we consider the following player's strategy. First, we assume that $T$ is an even number and we will address the situation where $T$ is odd later. The player plays 0 in the first half of the rounds. Let $W_1$ be the sum of numbers that the adversary plays in the first half of the rounds and $W_2$ be the sum in the second half. In other words, $W_1 = \sum_{t=1}^{T/2} w_t$ and $W_2 = \sum_{t=T/2+1}^{T} w_t$. In the second half of the rounds, the player plays $-\frac{W_1}{T/2}$. Since $|W_1| \leq T/2$, the player's choice $-\frac{W_1}{T/2}$ lies in $[-1, 1]$. The regret is equal to

$$W_2 \cdot \left(-\frac{W_1}{T/2}\right) + |W_1 + W_2|.$$

If $W_1 + W_2$ is non-negative, the regret equals $W_1 + W_2 - \frac{2W_1W_2}{T} = W_1 + W_2(1 - \frac{2W_1}{T}) \leq W_1 + \frac{T}{2}(1 - \frac{2W_1}{T}) = \frac{T}{2}$, where the inequality is because $1 - \frac{2W_1}{T} \geq 0$ and $W_2 \leq \frac{T}{2}$. If $W_1 + W_2$ is negative, the regret becomes $-W_1 - W_2 - \frac{2W_1W_2}{T} = -W_1 - W_2(1 + \frac{2W_1}{T}) \leq -W_1 + \frac{T}{2}(1 + \frac{2W_1}{T}) = \frac{T}{2}$, where the inequality is because $1 + \frac{2W_1}{T} \geq 0$ and $W_2 \geq -\frac{T}{2}$. Therefore, the regret is at most $\frac{T}{2}$. If

$T$ is odd, the player plays $0$ at the first round and the number of remaining rounds is $T-1$, which is even. The player then uses the previous strategy for an even $T$. In other words, the player plays $0$ from the first round to the $\frac{T+1}{2}$-th round and plays $-\frac{\sum_{t=2}^{(T+1)/2} w_t}{(T-1)/2}$ at all remaining rounds. The regret differs from the regret in the $(T-1)$-round game by at most $1$. Therefore, the regret is upper bounded by $\frac{T-1}{2} + 1 = \frac{T+1}{2} = \lceil \frac{T}{2} \rceil$. $\qquad\square$

The previous proposition demonstrated that the constant $\frac{1}{\sqrt{2}}$ could not be improved when $K = 2$, and thus could not be increased for an arbitrary $K$. In the next proposition, we show that our previous analysis of the fugal game was "tight" in a separate, asymptotic sense. When $K = o(T)$, the minimax regret of the fugal game relaxation is asymptotically (in $T$) equal to that of the original, switching-constrained OCO formulation. To understand the implication of this result, recall that the fugal game departed from the original game in two key ways. First, the player was permitted to choose non-discrete block lengths, $M_i \geq 0$, rather than only integral $M_i$. It is perhaps unsurprising that, as $T$ grows large, this restriction does not make a difference: intuitively, one can approximate $\frac{M_i}{T}$, where $M_i$ is non-integral and $T$ is small, arbitrarily well by $\frac{\tilde{M}_i}{\tilde{T}}$, where $\tilde{M}_i$ is integral but both it and $\tilde{T}$ are large. However, the fugal game also required the adversary to copy the player's switching pattern, and to play only $\pm 1$. It may be surprising that the combination of these various restrictions has no affect on the minimax rate, asymptotically.

To prove the result, we present a reduction which converts the player's optimal algorithm in achieving the fugal minimax rate, to an algorithm (Algorithm 1) for ordinary, switching-constrained OCO. The regret of this algorithm against an optimal adversary necessarily upper bounds the constrained OCO minimax rate by Appendix C. Intuitively, the player simulates a fugal game based on the real game, and chooses actions based on the simulated game. The player's strategy in Algorithm 1 "translates" in an appropriate manner from the actual game to a simulated fugal game, and proceeds according to the optimal strategy in the simulated game. In particular, she converts from the received, non-integral $w_i$ to an internal, stored set of fugal $w_i' \in \{\pm 1\}$, representing the closest approximation to a fugal game of the actual game. Once the adversary's cumulative action since the last switch, $W_t$, exceeds (in absolute value) the equivalent quantity in the fugal game, the player switches actions. She consults the fugal strategy as an oracle to pick which action to play, and the game continues. By some algebraic manipulation, we show that the regret of the "simulated" fugal game, and the real game, stay reasonably close. We can thus upper bound the ordinary minimax rate in terms of the fugal minimax rate and an additive term which disappears in the limit of $T$, obtaining the stated result.

**Proposition 29** (Asymptotic tightness of fugal lower bound). *For any fixed $K \geq 1$, we have the limit $\lim_{T \to \infty} \frac{1}{T} \mathcal{R}_{\mathrm{OCO}}(B^1, B^{*1}, K, T) = u_K(0)$, where $u_K(0)$ is defined in Lemma 16 and denotes the normalized minimax regret with no initial bias.*

*Proof.* Let $1 = m_1 < m_2 < \cdots < m_{K_T}$ denote all moving rounds. For any integer $1 < t \leq T$, let $p(t)$ be the largest integer such that $m_{p(t)} < t$. Recall the regret of a $T$-round fugal game with a maximum number of $k-1$ switches and no initial bias is given by

$$\sup_{\lambda > 0} \left( \sum_{i=1}^{k} M_i w_i x_i + \left| \sum_{i=1}^{k} M_i w_i \right| + \lambda \mathbb{1}[\sum_{i=1}^{k} M_i \neq T] \right).$$

Let $x_i^*(w_1, \ldots, w_{i-1}) : \{-1, 1\}^{i-1} \to [-1, 1]$ and $M_i^*(w_1, \ldots, w_i) : \{-1, 1\}^i \to \mathbb{R}_{\geq 0}$ be the optimal strategy of the player in the fugal game, where $i = 1, \ldots, K$. We will use this strategy to construct a player' strategy for the switching-constrained OCO, which is presented in Algorithm 1.

First, we claim that $K_T \leq K$. According to the algorithm, the instruction $K_t \leftarrow K_{t-1} + 1$ is executed when $t > 1$ and either $W_t \geq U_{p(t)}$ or $W_t \leq L_{p(t)}$ happens. In both cases, we have $|W_t| \geq M_{p(t)}^*(w_1', w_2', \ldots, w_{p(t)}')$. Since the $t$-th round is a moving round if the instruction $K_t \leftarrow K_{t-1} + 1$ is executed, we get $m_{p(t+1)} = t$. Since $|W_t| \leq t - 1 - m_{p(t)} + 1 = t - m_{p(t)} = m_{p(t+1)} - m_{p(t)}$, the inequality $m_{p(t+1)} - m_{p(t)} = m_{K_t} - m_{K_t - 1} \geq M_{p(t)}^*(w_1', w_2', \ldots, w_{p(t)}')$ must hold. Note that the above equality is true only if the $t$-th round is a moving round. Additionally, notice that for any $k$, $K_{m_k} = k$ and $p(m_k) = k - 1$. If $K_T \geq K + 1$, summing the inequality over all

**Algorithm 1:** Player's strategy for switching-constrained OCO derived from fugal games

**Output :** Player's moves $x_1, \ldots, x_T$.

**1** **for** $t = 1, \ldots, T$ **do**

**2** $\quad$ Observe $w_{t-1}$;

**3** $\quad$ **if** $t = 1$ **then**

**4** $\quad\quad$ $K_1 \leftarrow 1$;

**5** $\quad\quad$ Play $x_1 \leftarrow x_1^*$;

**6** $\quad$ **else**

**7** $\quad\quad$ $W_t \leftarrow \sum_{j=m_{p(t)}}^{t-1} w_j$;

**8** $\quad\quad$ $U_{p(t)} \leftarrow M_{p(t)}^*(w_1', w_2', \ldots, w_{p(t)-1}', 1)$;

**9** $\quad\quad$ $L_{p(t)} \leftarrow -M_{p(t)}^*(w_1', w_2', \ldots, w_{p(t)-1}', -1)$;

**10** $\quad\quad$ **if** $W_t \geq U_{p(t)}$ **then**

**11** $\quad\quad\quad$ $K_t \leftarrow K_{t-1} + 1$;

**12** $\quad\quad\quad$ $w_{p(t)}' \leftarrow 1$;

**13** $\quad\quad\quad$ Play $x_t \leftarrow x_{p(t)+1}^*(w_1', w_2', \ldots, w_{p(t)}')$;

**14** $\quad\quad$ **else if** $W_t \leq L_{p(t)}$ **then**

**15** $\quad\quad\quad$ $K_t \leftarrow K_{t-1} + 1$;

**16** $\quad\quad\quad$ $w_{p(t)}' \leftarrow -1$;

**17** $\quad\quad\quad$ Play $x_t \leftarrow x_{p(t)+1}^*(w_1', w_2', \ldots, w_{p(t)}')$;

**18** $\quad\quad$ **else**

**19** $\quad\quad\quad$ $K_t \leftarrow K_{t-1}$;

**20** $\quad\quad\quad$ Play $x_t \leftarrow x_{t-1}$;

**21** $\quad\quad$ **end**

**22** $\quad$ **end**

**23** **end**

---

$t \in \{m_k | 2 \leq k \leq K + 1\}$ yields

$$\sum_{k=2}^{K+1} (m_{K_{m_k}} - m_{K_{m_k}-1}) = \sum_{k=2}^{K+1} (m_k - m_{k-1}) = m_{K+1} - 1$$

$$\geq \sum_{k=2}^{K+1} M_{p(m_k)}^*(w_1', w_2', \ldots, w_{p(m_k)}') = \sum_{k=2}^{K+1} M_{k-1}^*(w_1', w_2', \ldots, w_{k-1}') = T,$$

where the last equality is because for any given sequence $w_1', w_2', \ldots, w_K'$, the sum $\sum_{k=1}^{K} M_k^*(w_1', w_2', \ldots, w_k')$ must be $T$. Since we assume $K_T \geq K + 1$, we deduce $T \geq m_{K_T} \geq m_{K+1} \geq T + 1$, which is a contradiction. Therefore, we establish $K_T \leq K$.

Since $K_T \leq K$, for the purpose of analysis, let us modify Line 1 and wait until $K_t = K + 1$. In other words, the algorithm terminates at the $T_0$-th round whenever $K_{T_0} = K + 1$ happens. We define $m_{K+1} = T_0$. The algorithm continues running even if $t > T$, provided that $K_t \leq K$. We define $T' = T_0 - 1 \geq T$. The $T'$-th round is the last round such that $K_{T'} = K$. Note that in the following calculations, $x_t$ and $w_t'$ refer to the assignments made in Algorithm 1. Since the adversary can always play 0 at the additional rounds (*i.e.*, $w_{T+1} = w_{T+2} = \cdots = w_{T'} = 0$), we have

$$\max_{w_1, \ldots, w_T} \sum_{t=1}^{T} x_t \cdot w_t + \left| \sum_{t=1}^{T} w_t \right| \leq \max_{w_1, \ldots, w_{T'}} \sum_{t=1}^{T'} x_t \cdot w_t + \left| \sum_{t=1}^{T'} w_t \right|$$

$$= \max_{w_1, \ldots, w_{T'}} \sum_{i=1}^{K} x_i^*(w_1', w_2', \ldots, w_{i-1}') \sum_{j=m_i}^{m_{i+1}-1} w_j + \left| \sum_{i=1}^{K} \sum_{j=m_i}^{m_{i+1}-1} w_j \right|.$$

For any $1 \leq i \leq K$, if $w_i' = 1$, since $|w_t| \leq 1$ for all $t$, we have $0 \leq \sum_{j=m_i}^{m_{i+1}-1} w_j - M_i^*(w_1', w_2', \ldots, w_i') \leq 1$. If $w_i' = -1$, similarly we get $0 \leq -\sum_{j=m_i}^{m_{i+1}-1} w_j -$

$M_i^*(w_1', w_2', \dots, w_i') \leq 1$. Combining these two cases gives $0 \leq w_i' \sum_{j=m_i}^{m_{i+1}-1} w_j - M_i^*(w_1', w_2', \dots, w_i') \leq 1$. Multiplying through by $w_i'$, we therefore obtain

$$\left| \sum_{j=m_i}^{m_{i+1}-1} w_j - w_i' M_i^*(w_1', w_2', \dots, w_i') \right| \leq 1 \,.$$

Thus the following upper bound holds

$$\mathcal{R}_{\text{OCO}}(B^1, B^{*1}, K, T)$$

$$\leq \max_{w_1, \dots, w_T} \sum_{t=1}^{T} x_t \cdot w_t + \left| \sum_{t=1}^{T} w_t \right|$$

$$\leq \max_{w_1, \dots, w_{T'}} \sum_{i=1}^{K} x_i^*(w_1', w_2', \dots, w_{i-1}') \sum_{j=m_i}^{m_{i+1}-1} w_j + \left| \sum_{i=1}^{K} \sum_{j=m_i}^{m_{i+1}-1} w_j \right|$$

$$\leq \max_{w_1, \dots, w_{T'}} \sum_{i=1}^{K} x_i^*(w_1', w_2', \dots, w_{i-1}') w_i' M_i^*(w_1', w_2', \dots, w_i') + \left| \sum_{i=1}^{K} w_i' M_i^*(w_1', w_2', \dots, w_i') \right| + 2K$$

$$= r_K(T, 0) + 2K \,,$$

where $r_K$ is defined in (2) and denotes the minimax regret of a $T$-round fugal game with a maximum number of $K - 1$ switches and no initial bias. Recalling Lemma 16 yields

$$\frac{1}{T} \mathcal{R}_{\text{OCO}}(B^1, B^{*1}, K, T) \leq \frac{1}{T} r_K(T, 0) + 2K = u_K(0) + \frac{2K}{T} \,.$$

Since the fugal game provides a lower bound for $\mathcal{R}_{\text{OCO}}(B^1, B^{*1}, K, T)$, it follows that

$$\frac{1}{T} \mathcal{R}_{\text{OCO}}(B^1, B^{*1}, K, T) \geq \frac{1}{T} r_K(T, 0) = u_K(0) \,.$$

By assumption, $\lim_{T \to \infty} \frac{2K}{T} = 0$. Thus the limit $\lim_{T \to \infty} \frac{1}{T} \mathcal{R}_{\text{OCO}}(B^1, B^{*1}, K, T)$ exists and equals $u_K(0)$. $\qquad\square$

# D  Additional Lower Bounds for Higher-Dimensional Switching-Constrained OCO

Proposition 30 proves a dimension-dependent lower bound. In other words, the problem is harder as the dimension becomes higher. The proof is based on Proposition 9 and the observation that if both the player and the adversary select from the $\infty$-norm unit ball, the problem can be decomposed into $n$ fully decoupled one-dimensional sub-problems.

**Proposition 30** (Lower bound for $\infty$-norm)**.** *The minimax regret $\mathcal{R}_{\text{OCO}}(B_\infty^n, B_\infty^{*n}, K, T)$ is at least $\frac{nT}{\sqrt{2K}}$.*

*Proof.* By (1), we have

$$\mathcal{R}_{\text{OCO}}(B_\infty^n, B_\infty^{*n}, K, T) = \inf_{\|x_1\|_\infty \leq 1} \sup_{\|w_1\|_\infty \leq 1} \dots \inf_{\|x_T\|_\infty \leq 1} \sup_{\|w_T\|_\infty \leq 1} \sup_{\lambda > 0}$$

$$\left( \sum_{i=1}^{T} w_i \cdot x_i + \left\| \sum_{i=1}^{T} w_i \right\|_1 + \lambda \mathbb{1}[c(x_1, \dots, x_T) \geq K] \right) \,.$$

Both terms are decomposable by coordinates as follows: the $j^{th}$ coordinate of $\sum_{i=1}^{T} w_i \cdot x_i = \sum_{j=1}^{n} w_{i,j} x_{i,j}$ and $\left\| \sum_{i=1}^{T} w_i \right\| = \sum_{j=1}^{n} \left| \sum_{i=1}^{T} w_{i,j} \right|$. Therefore by Proposition 9, we obtain

$$\mathcal{R}_{\text{OCO}}(B_\infty^n, B_\infty^{*n}, K, T) = n \mathcal{R}_{\text{OCO}}(B^1, B^{*1}, K, T) \geq \frac{nT}{\sqrt{2K}} \,.$$

$\qquad\square$

# E  Additional Upper Bounds for Switching-Constrained OCO

In this section, we derive upper bounds for switching-constrained OCO to match the lower bounds in Propositions 6 and 7. We begin with a simple algorithm achieving the correct minimax regret, $O(\frac{T}{\sqrt{K}})$, for any player's action set $\mathcal{D}$ and the function family $\mathcal{F}$ that the adversary chooses from.

**Proposition 2.** *If $\mathcal{D}$ is a convex and compact set from which the player draws $x_i$, and $\mathcal{F}$ is the family of differentiable convex functions on $\mathcal{D}$, with uniformly bounded gradient, from which the adversary chooses $f_i$, a mini-batching algorithm yields the upper bound $\mathcal{R}(T,K) \leq \lceil \frac{T}{K} \rceil O(\sqrt{K}) = O(\frac{T}{\sqrt{K}})$.*

*Proof.* First, we claim that the minimax regret $\mathcal{R}_{\text{OCO}}(\mathcal{D}, \mathcal{F}, K, T)$ is a non-decreasing function in $T$. To see this, consider the situation where we have more rounds. The adversary can play 0 in all additional rounds and this does not decrease the regret. Therefore, we obtain that $\mathcal{R}_{\text{OCO}}(\mathcal{D}, \mathcal{F}, K, T) \leq \mathcal{R}_{\text{OCO}}(\mathcal{D}, \mathcal{F}, K, T_1)$, where $T_1 = \lceil \frac{T}{K} \rceil K \geq T$.

In the sequel, we derive an upper bound for $\mathcal{R}_{\text{OCO}}(\mathcal{D}, \mathcal{F}, K, T_1)$. To attain the upper bound, we mini-batch the $T_1$ rounds into $K$ equisized epochs, each having size $\frac{T_1}{K} = \lceil \frac{T}{K} \rceil$. Let $E_i$ denote the set of all rounds that belong to the $i$-th epoch. We have $E_i = \{\frac{T_1}{K}(i-1)+1, \frac{T_1}{K}(i-1)+2, \ldots, \frac{T_1}{K}i\}$. The epoch loss of the $i$-th epoch is the average of loss functions in this epoch, *i.e.*, $\bar{f}_i \triangleq \frac{1}{|E_i|} \sum_{j \in E_i} f_j$. If we run a minimax optimal algorithm for unconstrained OCO (for example, online gradient descent [25]) on the epoch losses $\bar{f}_1, \ldots, \bar{f}_K$ and obtain the player's action sequence $\bar{x}_1, \ldots, \bar{x}_K$, our strategy is to play $\bar{x}_i$ at all rounds in the $i$-th epoch. This method was originally discussed in [6, 13]. Using this mini-batching method, we deduce that the regret is upper bounded by $\frac{T_1}{K} O(\sqrt{K}) = \lceil \frac{T}{K} \rceil O(\sqrt{K}) = O(\frac{T}{\sqrt{K}})$, where $O(\sqrt{K})$ is the standard upper bound of the regret of a $K$-round OCO game. $\square$

In the next two propositions, we seek a more precise understanding of the *exact* minimax rate — *i.e.* the constant in front of $\frac{T}{\sqrt{K}}$ — of switching-constrained online linear optimization, beginning with $n = 1$. In Appendix C, Proposition 4 demonstrated that we cannot hope to improve the constant in the lower bound, $\frac{1}{\sqrt{2}}$, for arbitrary $T$ and $K$. Further, Proposition 29 showed that the fugal game captures the correct constant, *asymptotically*. In the following proposition, we seek a direct, *non-asymptotic* bound on the constant in front of the one-dimensional minimax rate, $\tilde{O}(\frac{T}{\sqrt{K}})$. To do so, we more carefully examine the mini-batching technique from Proposition 2. We observe that it actually allows reuse of the exact *minimax* rate (including the constant) of vanilla unconstrained OCO, rather than simply algorithms like projected gradient descent in our original application of the technique.

**Lemma 31.** *If $T$ and $K$ are positive integers such that $T \geq K \geq 1$, the inequality $\lceil \frac{T}{K} \rceil \leq \frac{2T}{\sqrt{K(K+1)}}$ holds.*

*Proof.* If $T$ is divisible by $K$, we have $\lceil \frac{T}{K} \rceil = \frac{T}{K}$. Since $\sqrt{\frac{K+1}{K}} \leq 2$, we get $\frac{1}{K} \leq \frac{2}{\sqrt{K(K+1)}}$ and thus $\lceil \frac{T}{K} \rceil = \frac{T}{K} \leq \frac{2T}{\sqrt{K(K+1)}}$. In the sequel, we assume that $K$ cannot divide $T$. We consider the Euclidean division of $T$ by $K$. There exists unique positive integers $q$ and $r$ such that $T = qK + r$ and $1 \leq r \leq K$. The following inequality holds

$$\frac{q+1}{q+r/K} \leq \frac{q+1}{q+1/K} \leq \frac{2}{1+1/K} = 2 \cdot \frac{K}{K+1} \leq 2\sqrt{\frac{K}{K+1}}, \tag{14}$$

where the first inequality is because $\frac{q+1}{q+r/K}$ is a decreasing function in $r$ and the second inequality is because $\frac{q+1}{q+1/K}$ is a decreasing function in $q$. In light of (14), we have

$$\left\lceil \frac{T}{K} \right\rceil = q+1 \leq 2(q+r/K)\sqrt{\frac{K}{K+1}} = \frac{2(Kq+r)}{\sqrt{K(K+1)}} = \frac{2T}{\sqrt{K(K+1)}}.$$

$\square$

**Proposition 32** (Upper bound for 2-norm and dimension at least 2). *The minimax regret $\mathcal{R}_{\text{OCO}}(B_2^n, B_2^{*n}, K, T)$ satisfies $\mathcal{R}_{\text{OCO}}(B_2^n, B_2^{*n}, K, T) \leq \lceil \frac{T}{K} \rceil \sqrt{K} \leq \frac{2T}{\sqrt{K}}$ for all $n \geq 2$.*

*Proof.* Let $\mathcal{R}(T)$ denote the minimax regret of the vanilla $T$-round $n$-dimensional OCO without a switching constraint. It is defined by

$$\mathcal{R}(T) = \inf_{x_1 \in B_2^n} \sup_{w_1 \in B_2^{*n}} \cdots \inf_{x_T \in B_2^n} \sup_{w_T \in B_2^{*n}} \left( \sum_{i=1}^{T} w_i \cdot x_i + \left\| \sum_{i=1}^{T} w_i \right\| \right) .$$

Using the mini-batching argument that we used to show Proposition 2, we have $\mathcal{R}_{\text{OCO}}(B^n, B^{*n}, K, T) \leq \lceil \frac{T}{K} \rceil \mathcal{R}(K)$. [7] By Theorem 6 of Abernethy et al. [1], $\mathcal{R}(K) = \sqrt{K}$ when $n > 2$. In fact, when $n = 2$, the upper bound of Lemma 9 carries through, so $\mathcal{R}(K) = \sqrt{K}$ when $n = 2$ as well. Thus by Lemma 31, we have

$$\mathcal{R}_{\text{OCO}}(B_2^n, B_2^{*n}, K, T) \leq \left\lceil \frac{T}{K} \right\rceil \mathcal{R}(K) = \left\lceil \frac{T}{K} \right\rceil \sqrt{K} \leq \frac{2T}{\sqrt{K(K+1)}} \sqrt{K} = \frac{2T}{\sqrt{K+1}} < \frac{2T}{\sqrt{K}} .$$

$\square$

**Proposition 33.** *For any $p$ and $q$ such that $1 \leq p, q \leq \infty$, the minimax regret $\mathcal{R}_{\text{OCO}}(B_p^n, B_q^{*n}, K, T)$ is non-decreasing in the dimension $n$.*

*Proof.* We will show that for any $m < n$, it holds that $\mathcal{R}_{\text{OCO}}(B_p^m, B_q^{*m}, K, T) \leq \mathcal{R}_{\text{OCO}}(B_p^n, B_q^{*n}, K, T)$. We view $B_p^m$ ($B_q^{*m}$, respectively) as the subset of $B_p^n$ ($B_q^{*n}$, respectively) by setting the last $n - m$ coordinates to 0. Next, we show how to convert a minimax optimal player's strategy in the $n$-dimensional game into a player's strategy in the $m$-dimensional game. Let $x_i^*(x_1, w_1, \ldots, x_{i-1}, w_{i-1}) : B_p^n \times B_q^{*n} \times \cdots \times B_p^n \times B_q^{*n} \to B_p^n$ be the optimal strategy of the player in the $n$-dimensional game. Note that any adversary's choice $w_t \in B_q^{*m}$ can be viewed as a choice in $B_q^{*n}$. At the $t$-th round of the $m$-dimensional game, given the adversary's previous choices $w_1, \ldots, w_{t-1}$ and the player's previous choices $x_1, \ldots, x_{t-1}$, the player computes $x_t' = x_t^*(x_1, w_1, \ldots, x_{t-1}, w_{t-1})$ and plays $x_t = P(x_t')$, where $P$ is the orthogonal projection onto $B_p^m$ (*i.e.*, setting the last $n - m$ coordinates to 0). Notice that $w_t \cdot x_t = w_t \cdot x_t'$. Therefore, in light of (1), the regret of the $m$-dimensional game $\sum_{t=1}^{T} w_t \cdot x_t + \| \sum_{t=1}^{T} w_t \|_{p/(p-1)}$ equals the regret of the $n$-dimensional game $\sum_{t=1}^{T} w_t \cdot x_t' + \| \sum_{t=1}^{T} w_t \|_{p/(p-1)}$, and is thus at most $\mathcal{R}_{\text{OCO}}(B_p^n, B_q^{*n}, K, T)$. $\square$

**Proposition 34** (Upper bound for one dimension)**.** *The minimax regret $\mathcal{R}_{\text{OCO}}(B^1, B^{*1}, K, T)$ satisfies*

*(a) For all $K \geq 1$, $\mathcal{R}_{\text{OCO}}(B^1, B^{*1}, K, T) \leq \lceil \frac{T}{K} \rceil \min\{\sqrt{\frac{2(K+1)}{\pi}}, \sqrt{K}\} \leq 2\sqrt{\frac{2}{\pi}} \frac{T}{\sqrt{K}} < \frac{1.6T}{\sqrt{K}}$; and*

*(b) For all $K \geq 2$, $\mathcal{R}_{\text{OCO}}(B^1, B^{*1}, K, T) \leq \frac{\sqrt{3}}{2} \lceil \frac{T}{K} \rceil \sqrt{K} < 0.87 \lceil \frac{T}{K} \rceil \sqrt{K}$.*

*Proof.* As in Proposition 32, a more careful inspection of the mini-batching argument reveals that the minimax regret $\mathcal{R}_{\text{OCO}}(B^1, B^{*1}, K, T)$ is at most $\lceil \frac{T}{K} \rceil$ times $\mathcal{R}(K)$, the minimax rate of vanilla OCO. If $K$ is even, Theorem 10 of [21] implies that $\mathcal{R}(K) = \frac{K}{2^K} \binom{K}{\frac{K}{2}} \leq \sqrt{\frac{2K}{\pi}}$. McMahan and Abernethy [21] did not report the minimax regret if $K$ is odd. If $K$ is odd, according to (10) of [21], we have

$$\mathcal{R}(K) = \frac{1}{2^K} \sum_{i=0}^{K} \binom{K}{i} |2i - K| = \frac{4}{2^K} \sum_{i=0}^{(K-1)/2} \binom{K}{i} \left( \frac{K}{2} - i \right) .$$

The minuend equals

$$\sum_{i=0}^{(K-1)/2} \binom{K}{i} \frac{K}{2} = \frac{K}{2} \cdot \frac{2^K}{2} = \frac{K 2^K}{4} .$$

The subtrahend is given by

$$\sum_{i=0}^{(K-1)/2} \binom{K}{i} i = K \sum_{i=1}^{(K-1)/2} \binom{K-1}{i-1} = K \sum_{i=0}^{(K-3)/2} \binom{K-1}{i} = \frac{K}{2}\left(2^{K-1} - \binom{K-1}{\frac{K-1}{2}}\right).$$

Putting them together yields

$$\mathcal{R}(K) = \frac{K}{2^{K-1}}\binom{K-1}{\frac{K-1}{2}}.$$

Next, we verify that if $K$ is odd, $\mathcal{R}(K) = \mathcal{R}(K+1)$. We have

$$\begin{aligned}
\mathcal{R}(K+1) &= \frac{K+1}{2^{K+1}}\binom{K+1}{\frac{K+1}{2}} \\
&= \frac{K+1}{2^{K+1}} \cdot \frac{K+1}{\frac{K+1}{2}}\binom{K}{\frac{K-1}{2}} \\
&= \frac{K+1}{2^{K}}\binom{K}{\frac{K+1}{2}} \\
&= \frac{K+1}{2^{K}} \cdot \frac{K}{\frac{K+1}{2}}\binom{K-1}{\frac{K-1}{2}} \\
&= \mathcal{R}(K).
\end{aligned}$$

In other words, the regret $\mathcal{R}(K)$ obeys the following pattern

$$\mathcal{R}(1) = \mathcal{R}(2) < \mathcal{R}(3) = \mathcal{R}(4) < \cdots < \mathcal{R}(2n-1) = \mathcal{R}(2n) < \cdots .$$

Therefore, if $K$ is odd, it holds that

$$\mathcal{R}(K) = \mathcal{R}(K+1) \leq \sqrt{\frac{2(K+1)}{\pi}}.$$

As a result, for any $K$, even or odd, the following inequality holds

$$\mathcal{R}(K) \leq \sqrt{\frac{2(K+1)}{\pi}}.$$

By Lemma 31, we obtain

$$\begin{aligned}
\mathcal{R}_{\text{OCO}}(B^1, B^{*1}, K, T) &\leq \left\lceil \frac{T}{K} \right\rceil \mathcal{R}(K) \leq \left\lceil \frac{T}{K} \right\rceil \sqrt{\frac{2(K+1)}{\pi}} \\
&\leq \frac{2T}{\sqrt{K(K+1)}}\sqrt{\frac{2(K+1)}{\pi}} = 2\sqrt{\frac{2}{\pi}}\frac{T}{\sqrt{K}}.
\end{aligned}$$

Additionally, Proposition 33 and Proposition 32 imply

$$\mathcal{R}_{\text{OCO}}(B^1, B^{*1}, K, T) \leq \mathcal{R}_{\text{OCO}}(B_2^2, B_2^{*2}, K, T) \leq \left\lceil \frac{T}{K} \right\rceil \sqrt{K}.$$

As a consequence, we prove part (a)

$$\mathcal{R}_{\text{OCO}}(B^1, B^{*1}, K, T) \leq \left\lceil \frac{T}{K} \right\rceil \min\{\sqrt{\frac{2(K+1)}{\pi}}, \sqrt{K}\} \leq 2\sqrt{\frac{2}{\pi}}\frac{T}{\sqrt{K}} < \frac{1.6T}{\sqrt{K}}.$$

Next, we show part (b). Notice that if $K$ is odd,

$$\frac{\mathcal{R}(K+2)/\sqrt{K+2}}{\mathcal{R}(K)/\sqrt{K}} = \frac{\sqrt{K(K+2)}}{K+1} < 1.$$

Hence, for any odd $K \geq 3$, $\mathcal{R}(K)/\sqrt{K} \leq \mathcal{R}(3)/\sqrt{3} = \frac{\sqrt{3}}{2}$.

Recall that $\mathcal{R}(K)/\sqrt{K} \leq \sqrt{\frac{2}{\pi}}$ for when $K$ is even. Therefore, for all $K \geq 2$, we have

$$\mathcal{R}(K) \leq \frac{\sqrt{3}}{2}\sqrt{K} \leq 0.87\sqrt{K}\,.$$

Thus we obtain

$$\mathcal{R}_{\mathrm{OCO}}(B^1, B^{*1}, K, T) \leq \frac{\sqrt{3}}{2}\left\lceil \frac{T}{K} \right\rceil \sqrt{K} < 0.87\left\lceil \frac{T}{K} \right\rceil \sqrt{K}\,.$$

$\square$

We now easily show an upper bound on the minimax rate for the $\infty$-norm by the same idea.

**Proposition 35** (Upper bound for $\infty$-norm)**.** *The minimax regret $\mathcal{R}_{\mathrm{OCO}}(B_\infty^n, B_\infty^{*n}, K, T)$ is at most* $2\sqrt{\frac{2}{\pi}}\frac{nT}{\sqrt{K}}$.

*Proof.* The argument in Proposition 30 shows

$$\mathcal{R}_{\mathrm{OCO}}(B_\infty^n, B_\infty^{*n}, K, T) = n\mathcal{R}_{\mathrm{OCO}}(B^1, B^{*1}, K, T)\,.$$

The desired upper bound follows from $\mathcal{R}_{\mathrm{OCO}}(B^1, B^{*1}, K, T) \leq 2\sqrt{\frac{2}{\pi}}\frac{T}{\sqrt{K}}$ shown in Proposition 34.

$\square$

**Proposition 36** (Unequal block length for $K = 3$, informal)**.** *The minimax game between player and adversary when $K = 3$ has the player choose to make unequally spaced switches. In particular, the first switch happens at approximately $0.29T$ through the game, strictly before $0.33T$.*

*Proof.* The proof of Proposition 22 showed that $z_+$ and $z_-$ are $\pm(\sqrt{2}-1)$, via considering Case 2 of the proof and setting $i = 2$ and $z = 0$. Without loss of generality, we assume the optimal $z$ is $\sqrt{2} - 1$ ($z_+$ and $z_-$ are symmetric). Then, we translate the optimal $z$ into the location of the optimal first switch. Plugging $z' = \sqrt{2} - 1, z = 0, w = 1$ into the reparametrization $z'(t) = \frac{Tz+tw}{T-t}$ of Corollary 12, we get $t = (1 - \sqrt{2}/2)T \approx 0.29T$. Therefore, the optimal first switch happens at approximately $0.29T$.

$\square$

## Footnotes

[6]Intuitively, strengthening the player and weakening the adversary can only lower the minimax regret. Formally, this holds as a result of a few straightforward facts. If $X' \subseteq X$ and $Y' \subseteq Y$, then $\inf_{x \in X'} \sup_{y \in Y} f(x, y) \geq \inf_{x \in X} \sup_{y \in Y'} f(x, y)$. To apply this recursively, simply note that if for all $x \in X$ and $y \in Y$ we have $f(x, y) \leq g(x, y)$, then $\inf_{x \in X} \sup_{y \in Y} f(x, y) \leq \inf_{x \in X} \sup_{y \in Y} g(x, y)$. Thus restricting the range of possible values for each supremum term in $\mathcal{R}_{\mathrm{OCO}}(\mathcal{B}^n, \mathcal{L}^n, K, T)$ (corresponding to the adversary's choices), followed by enlarging the range of possible values for each infimum term (corresponding to the player's actions), only lowers the ultimate minimax regret.

[7]To be concrete, the minimax regret, $\mathcal{R}_{\text{OCO}}(B^1, B^{*1}, K, T)$ can only increase when we restrict the player to switch precisely every $\frac{T}{K}$ rounds. Then, conditioned on this player strategy, the regret term $\sum w_t x_t + |\sum w_t|$ is unchanged by forcing the adversary to also pick the same function on each $\frac{T}{K}$-sized block. Thus, $\frac{T}{K} \mathcal{R}(K)$ provides a valid upper bound as claimed.