[Reviews · NeurIPS 2020]

Review 1

Summary and Contributions: This paper deals with the problem of online convex optimization with hard switching costs: the online learning algorithm must not change the action more than K times. The authors focus on the case where the adversary selects only linear functions. The contribution of the paper is threefold: - A lower bound on the regret based on applying Abernethy's adversary (for the lower bound in regret in OCO) when the agent changes her action. - An upper bound on the regret based on taking batches of T/K consecutive epochs and applying stochastic gradient in the now K-horizon problem. This gives a matching bound on the order of K and T and exact for dimension n> 1 - An exact lower bound for n=1: this is obtained via a minmax analysis of a novel type of game ("fugal game").

Strengths: 1. This is the first paper to provide regret bounds (which also happen to be tight) with a constrained number of switches rather than just switching costs. 2. The fugal game (to obtain bounds for n =1 ) is a quite neat idea.

Weaknesses: 1. The techniques used to obtain the bounds for n > 1 are rather standard. It is also not clear if the fugal game can be extended/used for other cases.

Correctness: The paper is a theoretical one, and the proofs and claims seem correct.

Clarity: The paper is generally well written, however the notions of adaptive and oblivious adversary are not well explained before (it is unclear if the adaptive adversary chooses the function only or her action is dynamic as well) they are used in the introduction.

Relation to Prior Work: Work related to learning with switching costs is adequately discussed and compared with. However, there is a small issue related Cover's counterexample and the notion of regret. Cover's paper says that an adversary can force O(T) regret if she can change her action at each decision epoch; regret as defined before forces the adversary to pick a static action, so the above result does not apply.

Reproducibility: Yes

Additional Feedback: The authors mention that, in the studied case with continuous action spaces, no phase transition is observed; this is an interesting observation, however it would be nice to have some intuitive explanation why this is the case and what are the properties of this setting that allow this "no phase transition". ------- Update after the authors' reply --------- I would like to thank the authors for their clarifications, though it is still not clear if the fugal game can be used outside this specific case. Overall it is a solid paper with a neat result, my score remains the same.


Review 2

Summary and Contributions: This paper considers the problem of online convex optimization, in a scenario where the learning algorithm can only change its action K < T times. This is a natural scenario for many online decision-making problems. For instance if the current state x_i corresponds to the parameter settings of some software system and changing the action corresponds to pushing out a new update of the system, or if x_i corresponds to an allocation of an R&D budget within a company and changing the action means modifying how the R&D budget is allocated, then you would want to limit the number of times you need to do that. This type of problem has been considered extensively in the classic discrete-action combining-expert-advice scenario, and there is some past work in the OCO setting, but this paper gets tight minimax bounds. Interestingly, it shows that unlike the discrete case, in the OCO case there is no phase transition. Also it defines an interesting “fugue” game to tightly analyze the 1-dimensional case even up to constants. Overall, a good and substantial paper.

Strengths: The paper achieves tight minimax bounds for a natural online problem. It also shows that unlike in the discrete “combining expert advice” problem, there is no phase transition. Analysis is interesting and substantial.

Weaknesses: The space this is operating in is a little crowded - there are a variety of results over the years and they depend on specific assumptions of the problem setup.

Correctness: Yes, as far as I can tell

Clarity: Yes, the paper is well written.

Relation to Prior Work: Yes.

Reproducibility: Yes

Additional Feedback: Can you explain more what is the reason for the disappearance of the phase transition compared to the discrete game? In the discrete game without a switching bound, you would typically get a total cost of (1+epsilon)*OPT + (log n)/epsilon, and then set epsilon = sqrt((log n)/T) to balance the losses. This means that even without a switching bound, against an oblivious adversary, you’re switching between actions only about 1/sqrt(T) of the time anyway. On the other hand, you are constantly modifying your probability distribution every round (just that this change is only occasionally reflected in a switch of actions). In your game, there isn’t this distinction between the hidden probability distribution and the action played. Is that partly the difference? It would be great to have more discussion of this. Also, is there a fundamental difference between limiting the number of switches and having a cost per switch? I couldn’t get a sense of that from the discussion in appendix A. In relation to the Metrical Task System problem, my intuition is the “who goes first” aspect is minor, since once you have costs to switch, it doesn’t really hurt the adversary much to reveal its loss function first, so long as losses per round are small – the big difference is the quantity being compared against. Another (old) paper perhaps worth relating to is: Baruch Awerbuch, Yossi Azar, Amos Fiat, Frank Thomson Leighton: Making Commitments in the Face of Uncertainty: How to Pick a Winner Almost Every Time, STOC 1996, which analyzes limited switching from a multiplicative ratio perspective. Note added: Thanks to the authors for their informative response and clarifications.


Review 3

Summary and Contributions: This paper studies switching-constrained online convex optimization, where the learner can switch her action at most K times. The authors propose a mini-batch-paradigm-based online gradient descent algorithm for solving this problem, which is proved to enjoy an O(T/\sqrt{K}) regret bound. They also provide a matching lower bound for this problem.

Strengths: 1. The problem is interesting and well-motivated. Optimization under Constrained switching cost widely exists in real-world applications. It has been extensively studied in both learning with experts and the bandits setting, but rarely considered in the online convex optimization paradigm. 2. This paper successfully established the minimax bound for this problem, and proposed a mini-batching algorithm to achieve a matching upper bound.

Weaknesses: Novelty: 1. One of the main contribution of the paper is the algorithm proposed in Section 5 which achieves the minimax optimal regret bound. However, the main technique used here is the mini-batch paradigm which has been considered before, e.g., in [Dekel et al., 2011; Arora et al., 2012], and it seems that the proof technique is rather straightforward. 2. It seems that the adversary’s strategy and the minimax analysis in Section 4.1 are largely based on those of [Abernethy et al., 2008]. It would be better if the authors could add more discussions on what are the difficulties to adapt the minimax analysis for classic OCO into the switch-constrained setting. Significance: 2. The fugal game proposed in this paper is novel and very interesting, but I am unsure about the significance of this contribution since it is considered in the 1-d situation and it only improves the minimax bound by constant factors. --------------------------------------Post Rebuttal--------------------------- The authors cleared my concerns in the rebuttal. I am happy to raise my score.

Correctness: I have read the main paper and made high level checks of the proofs, and I didn’t find any significant errors.

Clarity: The paper is generally well-written and structured clearly.

Relation to Prior Work: The relation to prior work is clearly discussed.

Reproducibility: Yes

Additional Feedback:


Review 4

Summary and Contributions: The authors discuss the problem of OCO with limited switching. They prove minimax lower bounds and upper bounds, with sharp constants in one dimension and higher dimensions. They introduce a new novel approach, fugal games, to prove sharp constants for minimax lower bound of one dimensional problem.

Strengths: The theory is nicely presented, and the proofs are adequate and easy to follow (except the proofs for one-dimensional case, which I found very hard to follow and read; however, it is indeed the case when one tries to find a sharp constant for these types of problems). The contribution sits among other papers (such as "Consistent Online Learning: Convex and Submodular" and "Online learning over a finite action set with limited switching", and sheds light on the difference between convex set of actions and finite action set.

Weaknesses: There are no experiments, but I do not think it is a problem, as it is a theory paper, and I liked it that the authors decided to put parts of the proof in the main paper. The part of the paper discussing the relations with "Consistent Online Learning: Convex and Submodular" paper is not completely correct. What they prove in the paper is more general than that and the authors here decided to put a corrolary instead of the general result. However, this is understandable, as the main focus of this paper is minimax regret, not expected regret.

Correctness: As far as I read, and skimmed through parts of the appendix, the arguments seems to be correct. However, I did not read the one-dimensional case completely, as it was very technical and hard to follow.

Clarity: Yes, I enjoyed reading it.

Relation to Prior Work: Yes, as far as I know, the results are completely compared.

Reproducibility: Yes

Additional Feedback: I have read the authors' feedback, as well as other reviewers' comments, and still think that the score given is well adjusted.

[Author Response · NeurIPS 2020]

We are grateful to the reviewers for their detailed comments, for their judgment of the problem setting as "interesting
and well-motivated" (Reviewer 3), and for highlighting the novelty and intricacy of the fugal game (Reviewers 1-4). In
this rebuttal, we address general concerns shared across reviewers rather than responding to individual comments.

**Q1:** The proof techniques for the lower bound and mini-batching upper bound in dimension $n > 1$.

**A1:** As Reviewer 3 noted "there are a variety of results over the years", so we naturally built on and clearly acknowledged
prior work. Nonetheless, it was non-trivial to adapt prior techniques to the switching-constrained setting. First,
concerning the adversary's strategy (the "orthogonal trick", introduced by Abernethy et al.) for the lower bound: to
prove our result, we had to impose a certain switching pattern upon the adversary which was not obvious *a priori*,
since the adversary is free to play any functions they wish from round to round. Without constraining the adversary to
follow the player's switching pattern, the orthogonal trick cannot be applied. In addition, we found that this adversary's
strategy can be adapted for $n = 2$ - thereby avoiding the need for special treatment as in $n = 1$ - via a geometric fact
about the intersection of two closed half-spaces. Abernethy et al.'s original work only covered $n > 2$.

As the reviewers noted, the mini-batching algorithm of the upper bound was an existing innovation central to Arora et al.
2012, and we were careful to clearly cite this paper; we thank Reviewer 3 for bringing Dekel et al. 2011 to our attention,
and will add this reference. However, part of our appreciation of the result is the counter-intuitiveness that switching at
evenly sized intervals is optimal (up to a constant). This is surprising because the algorithm of Jaghargh et al. used a
Poisson process to decide when to switch actions and thus had unevenly sized blocks. Note that we also included more
technically involved results in the Appendix, including dimension-independent (Proposition 30) and more precise regret
(Proposition 34) upper bounds, which went beyond the immediate scope of mini-batching to attain tighter bounds.

More broadly, it is not uncommon in online optimization for minimax bounds in one-dimension to be technically
more demanding than for higher-dimensions, and our work is no different. However, we appreciate the elegance and
coherence with past work of our bounds for $n > 1$ and do not find their value diminished as a result; quite the contrary.

**Q2:** One dimensional lower bound and the broader applicability of fugal game.

**A2:** We should highlight that for $n = 1$ we provide an alternative, shorter, and also completely new lower in Proposition
7. However, the full machinery of the fugal game was necessary to achieve a bound asymptotically tight in $T$ (see
Proposition 8). Beyond our setting, the fugal game could be a valuable tool in proving lower bounds for any of the
many other natural switching-constrained continuous settings, including the convex bandit optimization and Gaussian
process bandit optimization, online submodular, and submodular bandit settings.

**Q3:** The intuition behind the lack of phase transition in the continuous, as opposed to discrete, setting

**A3:** We particularly thank Reviewer 2 for their excellent observation, which noted that in the discrete but not the
continuous setting, the player modifies their probability distribution each round. This observation is at the heart of the
difference in phase transition between the discrete and continuous settings. In our setting, such a randomization is
futile because *the adversary is stronger than in the discrete setting, and can choose the loss function $f_t$ based even
on the player's $t^{th}$ action.* (Intuitively, randomizing over a discrete set is similar to picking deterministically from its
convex hull.) By contrast, in prediction from experts, the adversary must be oblivious to avoid linear regret: they may
choose $f_t$ based only on $x_1, \ldots, x_{t-1}$ and knowledge of the player's randomized strategy. With a strong adversary,
any number of extra allotted switches can help the player, whereas with the weaker, oblivious adversary of prediction
from experts, an increased switching budget only aids the player up to a certain point. Since one usually assumes the
strongest adversary that yields sublinear regret, it is the continuity of the action space that permits an adaptive adversary
and thus the lack of a phase transition.

**Q4:** Cover's impossibility result.

**A4:** We thank Reviewer 1 for pointing out an error in our reference to Cover's impossibility result in Footnote 2. To
clarify, Cover showed that in the ordinary setting, any adaptive adversary can force linear regret; we state this on page 2.
The analogous impossibility result with a switching constraint was proven by Altschuler et al., which we cite in the next
paragraph. The footnote should have referred to this analogous fact instead; we will correct this in the revision.

**Q5:** The difference between the switching-constrained and switching-cost settings.

**A5:** In the discrete setting, there is indeed a duality between the two: their minimax rates are within a polylogarithmic
factor in $T$ for certain regimes (see Altschuler et al.). We have not previously considered the switching-cost formulation,
but of course a $K$-budgeted mini-batching algorithm achieves switching-cost penalty $O(T/\sqrt{K}) + cK$; optimally
setting $K = \left(T/2c\right)^{2/3}$ yields penalty $O(T^{2/3}c^{1/3})$. It is not immediately clear whether this is optimal for certain $c$ or
if there is a reduction in the other direction, so we defer this interesting topic to future work. We also thank Reviewer 2
for bringing the Awerbuch et al. paper to our attention, and we will add it to the Related Work in the revision.

[Meta-Review · NeurIPS 2020]

The initial four reviews recommended accepting, based on the strength of the results and the interesting ideas. However, some issues were pointed out mainly related to the novelty value of the work and relation to several earlier results in the field. The authors in their reply provided an informative discussion on these issues. This satisfied the reviewers and strengthened their confidence for accepting the paper.